# Approximation Rates and VC-Dimension Bounds for (P)ReLU MLP Mixture of Experts

**Anastasis Kratsios**                                           *kratsioa@mcmaster.ca*
*Vector Institute and McMaster University, Canada*

**Haitz Sáez de Ocáriz Borde**                                  *chri6704@ox.ac.uk*
*Oxford University, United Kingdom*

**Takashi Furuya**                                              *takashi.furuya0101@gmail.com*
*Shimane University, Japan*

**Marc T. Law**                                                 *marcl@nvidia.com*
*NVIDIA, Canada*

Reviewed on OpenReview: *https://openreview.net/forum?id=oeg2ncuSPz*

## Abstract

Mixture-of-Experts (MoEs) can scale up beyond traditional deep learning models by employing a routing strategy in which each input is processed by a single "expert" deep learning model. This strategy allows us to scale up the number of parameters defining the MoE while maintaining sparse activation, i.e., MoEs only load a small number of their total parameters into GPU VRAM for the forward pass depending on the input. In this paper, we provide an approximation and learning-theoretic analysis of mixtures of expert MultiLayer Perceptrons (MLPs) with (P)ReLU activation functions. We first prove that for every error level $\varepsilon > 0$ and every Lipschitz function $f : [0,1]^n \to \mathbb{R}$, one can construct a MoMLP model (a Mixture-of-Experts comprising of (P)ReLU MLPs) which uniformly approximates $f$ to $\varepsilon$ accuracy over $[0,1]^n$, while only requiring networks of $\mathcal{O}(\varepsilon^{-1})$ parameters to be loaded in memory. Additionally, we show that MoMLPs can generalize since the entire MoMLP model has a (finite) VC dimension of $\tilde{O}(L \max\{nL, JW\})$, if there are $L$ experts and each expert has a depth and width of $J$ and $W$, respectively.

## 1 Introduction

With the advent of large foundation models, scaling deep learning models beyond the capacity of a single machine has become increasingly important. Mixture of Experts (MoE) models offer a solution to this challenge through a sparse activation strategy. In MoEs, each input is first *routed* to one of many *expert* deep learning models and then processed by that expert. This approach allows MoEs to scale effectively while maintaining a low or constant computational cost during the forward pass, as only a subset of the overall model needs to be loaded into GPU video random-access memory (VRAM) for a given input. This has led to MoEs such as Mixtral (Jiang et al., 2024), Gemini (Google, 2023), and several others, e.g. (Jacobs et al., 1991; Jordan & Xu, 1995; Meila & Jordan, 2000; Shazeer et al., 2017; Guu et al., 2020; Lepikhin et al., 2021; Fedus et al., 2022; Barham et al., 2022; Majid & Tudisco, 2024), to become a viable solution in scaling up large language models (Radford et al., 2018; Brown et al., 2020). However, the analytical and statistical foundations of MoEs in deep learning are less understood compared to their empirical investigations.

This paper adds to the theoretical understanding of this subject by studying MoEs whose experts are (small) multilayer perceptrons (MLPs) with (P)ReLU activation function (MoMLPs). A key feature of MoEs is that they can maintain a small/fixed computational cost during the forward pass, for any given input $x \in [0,1]^n$,

even if the overall model complexity may be large. Our main result (Theorem 4.1) analyzes the complexity of MoEs when uniformly approximating an arbitrary Lipschitz (Lebesgue) almost-everywhere continuously differentiable function $f : [0, 1]^n \to \mathbb{R}^m$ by an MoMLP with (P)ReLU activation function to any prespecified error $\varepsilon > 0$. We focus on the trade-off between the maximum number of parameters loaded into VRAM by any expert model $\{\hat{f}_l\}_{l=1}^\ell$ while predicting from any given input, against the total number of experts required to maintain that constant number of activated parameters in the forward pass. Summarized in Table 1, our main result shows that a constant *active complexity* in the forward pass can be maintained among all experts, but at the cost of an exponentially large number of locally-specialized experts $\{\hat{f}\}_{l=1}^\ell$ and regions of specialization $\{C_l\}_{l=1}^\ell$. Our complexity estimates are approximately optimal as they nearly match the Vapnik-Chervonenkis (VC) lower bounds derived in Shen et al. (2021). That is, the uniform approximation of an arbitrary such $f$ on $[0, 1]^n$, with an an error of $\varepsilon > 0$, requires at least $\Omega(\varepsilon^{-n/2})$ *total model parameters* (Yarotsky, 2018; Kratsios & Papon, 2022; Shen et al., 2021; 2022b). It is here where MoEs have an advantage since not all of their parameters need to be loaded into active memory for any given input; thus, MoEs are genuinely sparsely activated.

It is important to note that the results recorded in Table 1 represent the *worst-case* scenario, meaning they pertain to the most challenging target function within the class of $\alpha$-Hölder functions. Approximating such a function necessarily requires a large number of experts in the MoE to achieve the desired accuracy, similar to how very large MLPs would be theoretically required. Nevertheless, in practice, one typically does not encounter worst-case functions. MoEs can still perform well even with a small number of experts in the mixture model and relatively few parameters per expert. It is worth noting, however, that analyzing non-worst-case scenarios is an interesting research question, separate from the worst-case analysis addressed in this manuscript.

From the statistical learning perspective, a key property of MoEs (e.g. the top MoMLP model) is that they can maintain a given level of activation in the forward pass while the entire MoE model can maintain a finite VC-dimension (Theorem 4.2). This is key, for instance, in classification applications, as the fundamental theorem of PAC learning (see e.g. (Shalev-Shwartz & Ben-David, 2014, Theorem 6.7) or the results of Blumer et al. (1989); Hanneke (2016); Brukhim et al. (2022)) implies that such a machine learning model generalizes beyond the training data if and only if it has finite VC dimension.

**Summary of Contributions**    Table 1 summarizes our main contributions, both to the approximation theory and learning theory of MoE models, in the context of the toy mixture of (P)ReLU MLP models. All results illustrate the trade-off between individual (expert) complexity and the complexity shared across the set of experts when uniformly approximating a target $\alpha$-Hölder function; $0 < \alpha \leq 1$.

Our *approximation* theorem (Theorem 4.1) records the number of parameters required to perform a uniform approximation on a high-dimensional Euclidean space on $[0, 1]^n$. The first result juxtaposes the complexity of *each expert (PReLU MLP)* against the total number of experts required to achieve a given level of approximation accuracy. The user controls the number of experts vs. the complexity of each expert using a hyperparameter $r \in \mathbb{R}$. As shown in Table 1. Small values of $r < 0$ encode the "few large experts regime", whereas large values of $0 \leq r$ capture the "many small experts regime".

Our *statistical learning* guarantee result (Theorem 4.2) yields a bound on the VC dimension of the entire class of MoEs with just enough approximation power to perform this approximation. As summarized in Table 1, the result quantitatively shows the degradation of model generalization as the number of experts increases; i.e. $r$ becomes large.

Observe that, setting $r = \frac{2}{n}$ in Table 1, yields for ReLU MLPs derived in Yarotsky (2017); the optimality of which is expressed in terms of VC dimension in Shen et al. (2022b). Likewise, the VC dimension of the MoMLP is roughly equal to that of ReLU MLPs computed in Bartlett et al. (2019).

## 2    Related Work

**Deep Learning Models with Few Parameters in Active Memory.**    Deep learning models with highly oscillatory "super-expressive" activation functions (Yarotsky & Zhevnerchuk, 2020; Yarotsky, 2021;

*Table 1:* **No. Parameters and VC-Dimension** of MoMLP with *no. experts-to-expert-complexity parameter* $r \in (-\infty, \frac{2}{n}]$; performing an $0 < \varepsilon \leq 1$ approximation of an $\alpha$-Hölder function $f : [0,1]^n \to \mathbb{R}$; $n \in \mathbb{N}$. When $r \geq 0$ more model complexity is distributed across many "small experts". When $r < 0$, fewer experts define the MoE and, as a result, each expert MLP must depend on more parameters such that the entire MoE obtain an accurate approximation of the target function. We also record the total number of parameters defining the MoMLP, including those which are not loaded in the forward pass but can be stored offline.

| Parameter | Estimate |
|---|---|
| No. Parameters Per Expert | $\mathcal{O}(\max\{1, \varepsilon^{-r}\})$ |
| No. Experts | $\mathcal{O}\big(\max\{1, \varepsilon^{2r/n - 1/\alpha}\}\big)$ |
| Parameters MoMLP (Offline) | $\mathcal{O}\big(\max\{1, \varepsilon^{-r}\} \max\{1, \varepsilon^{2r/n - 1/\alpha}\}\big)$ |
| VC Dimension MoE | $\tilde{\mathcal{O}}\big(\max\{1, \varepsilon^{2r/n - 1/\alpha}\} \max\{\varepsilon^{2r/n - 1/\alpha}, \varepsilon^{-r}\}\big)$ |

Zhang et al., 2022) are known to achieve dimension-free approximation rates, thus effectively require a (relatively) feasible number of parameters to be loaded into VRAM during the forward pass. As we will see in Proposition 4.4, many of these models have an infinite VC dimension even when they are restricted to having a bounded depth and width; see Jiao et al. (2023, Lemma 3.1) for ReLU-Sin-2ˣ-networks. Their unbounded VC dimension implies that the classifiers implemented by these models do not generalize on classification problems. Thus, the real performance of these models does not need to achieve the approximation-theoretic optima since they can only learn from a finite number of noisy training instances. Alternatively, a feasible number of parameters in deep learning models with standard activation functions may be guaranteed by restricting classes of well-behaved target functions such as Barron functions (Barron, 1993), functions of mixed-smoothness (Suzuki, 2018), highly smooth functions (Mhaskar, 1996; Galimberti et al., 2022; Gonon et al., 2023; Opschoor et al., 2022), convex functions (Bach, 2017), functions with compositional structure (Mhaskar et al., 2017), or other restricted classes. However, there are generally no guarantees that a target function encountered in practice has the necessary structure for these desired approximation theorems to hold.

**Universal Approximation in Deep Learning.** Several results have recently considered the expression power of deep learning models. These include universal approximation guarantees for MLPs (Cybenko, 1989; Hornik et al., 1989; Lu et al., 2017; Suzuki, 2018; Yarotsky, 2017; 2018; Voigtlaender & Petersen, 2019; Bolcskei et al., 2019; Gühring et al., 2020; De Ryck et al., 2021; DeVore et al., 2021; Daubechies et al., 2022; Kratsios & Zamanlooy, 2022; Zhang et al., 2022; Opschoor et al., 2022; Zamanlooy & Kratsios, 2022; Shen et al., 2022b; Cuchiero et al., 2023; Voigtlaender, 2023; Benth et al., 2023; Mao & Zhou, 2023; Yang & Zhou, 2024), CNNs (Petersen & Voigtlaender, 2020; Yarotsky, 2022), spiking neural networks (Neuman & Petersen, 2024), residual neural networks (Tabuada & Gharesifard, 2021), transformers (Yun et al., 2019; 2020; Kratsios & Papon, 2022; Fang et al., 2023), random neural networks (Gonon et al., 2023), recurrent neural network models (Grigoryeva & Ortega, 2018; Gonon & Ortega, 2021; Hutter et al., 2022; Galimberti et al., 2022; hoon Song et al., 2023), and several others. In each these cases, one typically considers the expressivity of a single "expert" model and not a mixture thereof. Our analysis can be customized to any of these settings to yield analogues of Theorem 4.1.

**Foundations of MoEs.** MoE models have been heavily studied since their inception. Most results have focused on identifying the correct expert to best route any given input to (Teicher, 1960; 1963; Wang et al., 1996), the construction of effective routing mechanisms (Wang et al., 2017) selection (Wang et al., 1996), MoE training (Larochelle et al., 2009; Akbari et al., 2024), statistical convergence guarantees for classes of MoEs (Chen, 1995; Ho et al., 2022), robustness guarantees for such models (Puigcerver et al., 2022), amongst several other types of guarantees. However, to our knowledge, there are no available approximation guarantees for MoE or VC-dimension bounds for deep-learning-based MoEs. Thus, our results would be adding to the approximation theoretic foundations of MoE models as well as to the statistical foundations of deep-learning-based MoEs.

**Prototypes and Partitioning.** Each region in our learned partition of the input space is associated with a *representative point* therein called a *prototype*. Prototypes (also called *landmarks*) are routinely used in image classification (Mensink et al., 2012), few-shot learning (Snell et al., 2017; Cao et al., 2020), dimensionality reduction (Law et al., 2019), in complex networks (Keller-Ressel & Nargang, 2023), and geometric deep learning (Ghadimi Atigh et al., 2021) to tractably encode massive structures. They are also standard in classical clustering algorithms such as $K$-medoids or $K$-means, wherein the part associated with each medoid (resp. centroid) defines a Voronoi cell or Voronoi region (Voronoi, 1908). Moreover, while partitioning is commonly employed in deep learning for various purposes, such as proving universal approximation theorems (Yarotsky, 2017; Lu et al., 2021b; Gühring & Raslan, 2021) or facilitating clustering-based learning (Zamanlooy & Kratsios, 2022; Trask et al., 2022; Ali & Nouy, 2023; Srivastava et al., 2022), existing approaches typically involve loading the entire model into VRAM. Our approach, however, differs by relying on a learned partition of the input space, where each part is associated with a distinct small neural network. Importantly, the complete set of networks forming the MoMLPs does not need to be simultaneously loaded into VRAM during training or inference.

**Paper Overview.** Our paper is organized as follows. Section 3 contains preliminary notation, definitions, and mathematical background required for the formulation of our main results. Section 4 contains our main approximation (Theorem 4.1) and learning theoretic (Theorem 4.2) guarantees. Section 5 dives into the details of why *MoEs can achieve arbitrary precision while maintaining a feasible active computational complexity* by explaining the derivation of our main approximation theorem; the details of which are relegated to Appendix B. A technical version (Theorem 5.3) of our main approximation guarantee is then presented, which allows for the approximation of continuous functions of arbitrarily low regularity and for the organization of the experts defining the MoMLP via a decision tree implementing the indicator function to a Voronoi diagram of $[9,1]^d$. Section 6 contains technical derivations of our main results as well as experimental elucidation of the benefit of MoEs, and specifically the toy MoMLP model.

## 3 Preliminaries

We standardize our notation, define the necessary mathematical formalisms to state our main results and define our toy MoE Model.

**Notation** We use the following notation: for any $f, g : \mathbb{R} \to \mathbb{R}$, we write $f \in \mathcal{O}(g)$ if there exist $x_0 \in \mathbb{R}$ and $M \geq 0$ such that for each $x \geq x_0$ we have $|f(x)| \leq Mg(x_0)$. Similarly, we write $f \in \Omega(g)$ to denote the relation $g \in O(f)$. The ReLU *activation function* is given for every $x \in \mathbb{R}$ by $\mathrm{ReLU}(x) = \max\{x, 0\}$. For each $n \in \mathbb{N}_+$ and $C \subseteq \mathbb{R}^n$, the *indicator function* $I_C$ of $C$ is defined by: for each $x \in \mathbb{R}^n$ set $I_C(x) = 1$ if $x \in C$ and is 0 otherwise.

### 3.1 Background

**Multi-Layer Perceptrons** We will consider MLPs with trainable PReLU activation functions.

**Definition 3.1** (Trainable PReLU)**.** We define the trainable PReLU activation function $\sigma : \mathbb{R} \times \mathbb{R} \to \mathbb{R}$ for each input $x \in \mathbb{R}$ and each parameter $\gamma \in \mathbb{R}$ as follows:

$$\sigma_\gamma(x) \overset{\text{def.}}{=} \sigma(x, \gamma) \overset{\text{def.}}{=} \begin{cases} x & \text{if } x \geq 0, \\ \gamma x & \text{otherwise.} \end{cases}$$

PReLU generalizes ReLU since $\mathrm{ReLU}(x) = \sigma_0(x)$, and it makes the hyperparameter $\gamma$ of a Leaky ReLU learnable. We will often be applying our trainable activation functions component-wise. For positive integers $n, m$, we denote the set of $n \times m$ matrices by $\mathbb{R}^{n \times m}$. More precisely, we mean the following operation defined for any $N \in \mathbb{N}$, $\bar{\gamma} \in \mathbb{R}^N$ with $i^{th}$ entry denoted as $\bar{\gamma}_i$, and $x \in \mathbb{R}^N$, by

$$\sigma_{\bar{\gamma}} \bullet x \overset{\text{def.}}{=} \left( \sigma_{\bar{\gamma}_i}(x_i) \right)_{i=1}^N.$$

We now define the class of multilayer perceptions (MLPs), with trainable activation functions. Fix $J \in \mathbb{N}$ and a multi-index $[d] \stackrel{\text{def.}}{=} (d_0, \ldots, d_{J+1})$, and let $P([d]) = \sum_{j=0}^{J} d_j(d_{j+1} + 2)$. We identify any vector $\theta \in \mathbb{R}^{P([d])}$ with

$$\theta \leftrightarrow \left(A^{(j)}, b^{(j)}, \bar{\gamma}^{(j)}\right)_{j=0}^{J} \text{ and } \left(A^{(j)}, b^{(j)}, \bar{\gamma}^{(j)}\right) \in \mathbb{R}^{d_{j+1} \times d_j} \times \mathbb{R}^{d_j} \times \mathbb{R}^{d_j}. \tag{1}$$

We recursively define the representation function of a $[d]$-dimensional network by

$$\begin{aligned}
\mathbb{R}^{P([d])} \times \mathbb{R}^{d_0} \ni (\theta, x) &\mapsto \hat{f}_\theta(x) \stackrel{\text{def.}}{=} A^{(J)} x^{(J)} + b^{(J)}, \\
x^{(j+1)} &\stackrel{\text{def.}}{=} \sigma_{\bar{\gamma}^{(j)}} \bullet (A^{(j)} x^{(j)} + b^{(j)}) \qquad \text{for } j = 0, \ldots, J-1 \\
x^{(0)} &\stackrel{\text{def.}}{=} x.
\end{aligned} \tag{2}$$

We denote by $\mathcal{NN}_{[d]}^\sigma$ the family of $[d]$-dimensional *multilayer perceptrons* (MLPs), $\{\hat{f}_\theta\}_{\theta \in \mathbb{R}^{P([d])}}$ described by equation 2. The subset of $\mathcal{NN}_{[d]}^\sigma$ consisting of networks $\hat{f}_\theta$ with each $\bar{\gamma}_i^{(j)} = (1, 0)$ in equation 2 is denoted by $\mathcal{NN}_{[d]}^{\text{ReLU}}$ and consists of the familiar deep ReLU MLPs. The set of ReLU MLPs with *depth* $J$ and *width* $W$ is denoted by $\mathcal{NN}_{J,W:n,m}^\sigma = \cup_{[d]} \mathcal{NN}_{[d]}^\sigma$, where the union is taken over all multi-indices $[d] = [d_0, \ldots, d_{\tilde{J}}]$ with $n = d_0$, $m = d_{J+1}$, $d_0, \ldots, d_{J+1} \leq W$, and $\tilde{J} \leq J$.

**VC dimension** Let $\mathcal{F}$ be a set of functions from a subset $\mathcal{X} \subseteq \mathbb{R}^n$ to $\{0, 1\}$; i.e. binary classifiers on $\mathcal{X}$. The set $\mathcal{F}$ shatters (in the classical sense) a $k$-point subset $\{x_i\}_{i=1}^k \subseteq \mathcal{X}$ if $\mathcal{F}$ can represent every possible set of labels on those $k$-points; i.e. if $\#\{(\hat{f}(x_i))_{i=1}^k \in \{0, 1\}^k : \hat{f} \in \mathcal{F}\} = 2^k$.

As in Shen et al. (2022b), we extend the definition of shattering from binary classifiers to real-valued functions as follows. Let $\mathcal{F}$ be a set of functions from $[0, 1]^n$ to $\mathbb{R}$. The set $\mathcal{F}$ is said to shatter a $k$-point set $\{x_i\}_{i=1}^k \subseteq \mathcal{X}$ if

$$\{I_{(0,\infty)} \circ f : f \in \mathcal{F}\} \tag{3}$$

shatters it, i.e. if all possible classifiers on $\{x_i\}_{i=1}^k$ are implementable in the sense that $\{I_{(0,\infty)} \circ f : f \in \mathcal{F}\} = \{0, 1\}^{\{x_i\}_{i=1}^k}$; here $I_{(0,\infty)}(t) = 1$ if $t > 0$ and equals to 0 otherwise. Denoted by $\text{VC}(\mathcal{F})$, the *VC dimension* of $\mathcal{F}$ is the cardinality of the largest $k$-point subset shattered by $\mathcal{F}$. If $k$ is unbounded, then we say that $\mathcal{F}$ has an infinite VC dimension (over $\mathcal{X}$). One can show, see Bartlett et al. (2019), that the VC-dimension of any such $\mathcal{F}$ is roughly the same as the pseudo-dimension of Pollard (1990) for a small modification of $\mathcal{F}$.

VC dimension measures the richness of a class of functions. For example, in Harvey et al. (2017, Theorem 1), the authors showed that the set of MLPs with ReLU activation function with $L \in \mathbb{N}_+$ layers, width and $W \in \mathbb{N}_+$ satisfying $W > O(L) > C^2$, where $C \geq 640$, satisfies

$$\text{VC}(\mathcal{NN}_{W,L}^\sigma) \in \Omega\left(WL \log_2(W/L)\right). \tag{4}$$

Nearly matching upper bounds are in Bartlett et al. (2019).

**Definition: Our Toy Mixture of Experts Model** We study the following toy MoE model, where each expert (P)ReLU MLP specializes in a distinct region of the input domain $[0, 1]^n$. Informally, these regions $C_1, \ldots, C_\ell$ correspond to the sets of closest points (Voronoi cells) from a finite set of prototypes/landmarks $p_1, \ldots, p_\ell$ in $[0, 1]^n$, as illustrated in Figure 1. Associated to each region $C_i$ is a single expert MLP $\hat{f}_i$ with (P)ReLU activation function responsible for approximating the

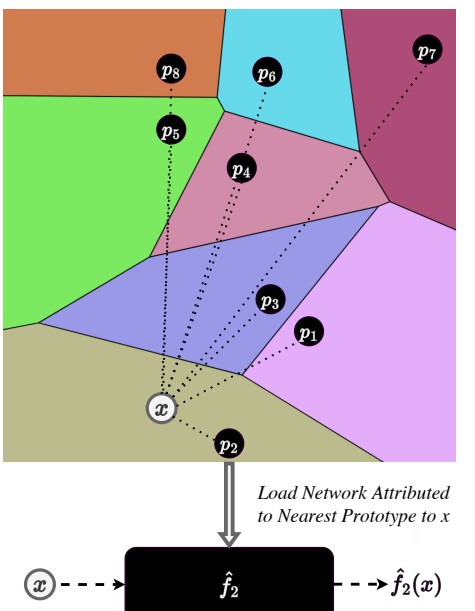

*Figure 1:* 1) The distance from each input $x$ to all prototypes $p_1, \ldots, p_8$ ($\ell = 8$) is queried. 2) The network ($\hat{f}_2$ in the figure) assigned to the nearest prototype ($p_2$), is loaded onto the GPU and used for prediction.

target function *only thereon.* Here, the sparse gating procedure which *routes* any given input $x \in [0,1]^n$ to the expert corresponding to the nearest prototype $p_i$ is implemented by a (finite) *routing tree* $\mathcal{T} = (V, E)$ whose nodes $V$ are points in $[0,1]^n$ and leaves (terminal nodes) are the points $p_1, \ldots, p_\ell$.

We now formally define the classes of MoMLPs.

**Definition 3.2** (MoMLPs). Let $J, W, L, n \in \mathbb{N}$ and fix an activation function $\sigma \in C(\mathbb{R})$. The set of MoMLPs with at-most $L$ leaves, depth $J$, and width $W$, denoted by $\mathcal{NP}^\sigma_{J,W,L:n,m}$, consists of all functions $\hat{f} : \mathbb{R}^n \to \mathbb{R}^m$ satisfying $\hat{f} = \sum_{i=1}^{L} f_i I_{C_i}$ where for $f_1, \ldots, f_L \in \mathcal{NN}^{\text{PReLU}}_{J,W}$, and *distinct* prototypes $p_1, \ldots, p_L \in \mathbb{R}^n$; inducing the Voronoi cells $\{C_i\}_{i=1}^{L}$ where

$$C_i \stackrel{\text{def.}}{=} \tilde{C}_i \setminus \bigcup_{j<i} \tilde{C}_j, \tag{5}$$

where for $i = 1, \ldots, L$ the (non-disjoint) cells are

$$\tilde{C}_i \stackrel{\text{def.}}{=} \left\{ x \in [0,1]^n : \|x - p_i\| = \min_{j \in \{1,\ldots,L\}} \|x - p_j\| \right\}. \tag{6}$$

**The Routing Trees**  The structure in any MoMLP is any tree with root node $\mathbb{R}^n$ and leaves given by the pairs $\{(p_i, f_i)\}_{i=1}^{L}$ or equivalently $\{(C_i, f_i)\}_{i=1}^{L}$. The purpose of any such tree is simply to efficiently route an input $x \in \mathbb{R}^n$ to one of the $L$ "leaf networks" (the experts) $f_1, \ldots, f_L$ using $\mathcal{O}(\log(L))$ queries; to identify which Voronoi cells $\{C_i\}_{i=1}^{L}$ the point $x$ is contained in. We leave the precise set of queries executed by the routing tree $\mathcal{T}$ abstract so as to allow for maximal design freedom. However, we do ask that $\mathcal{T}$ encodes a decision tree executing a sequence of queries at each node along a branch which implements the following function, routing any $x \in [0,1]^d$ to its nearest cell in the disjoint Voronoi cells $\{\tilde{C}_l\}$; i.e. $\mathcal{T}$ implements

$$\mathbb{R}^d \ni x \mapsto \sum_{l=1}^{L} l \, I_{x \in C_l} \in [L]. \tag{7}$$

*Example* 1 (Toy Implementation of equation 7 on Real Line). Set $d = 1$, consider the prototypes $\{1/8, 3/8, 5/8, 7/8\}$, and queries $q_{1,1}(x) \stackrel{\text{def.}}{=} I(|x - 1/4| < |x - 3/4|)$, $q_{2,1}(x) \stackrel{\text{def.}}{=} I(|x - 1/8| < |x - 3/8|)$, and $q_{2,2} \stackrel{\text{def.}}{=} I(|x - 5/8| < |x - 7/8|)$. The decision tree in Algorithm 1 implements equation 7.

*Remark* 3.3 (Partitioning in equation 5 in classical computer science). The partitioning technique used to define equation 5 is standard, see e.g. Krauthgamer et al. (2005, Proof of Lemma 1.7; page 846). It is employed to ensure disjointness of the Voronoi cells; this guarantees that no input is assigned to multiple prototypes. To keep notation tidy, we use $\mathcal{NP}^\sigma_{J,W,L}$ (resp. $\mathcal{N}^\sigma_{W,L}$), to abbreviate $\mathcal{NP}^\sigma_{J,W,L:n,m}$ (resp. $\mathcal{N}^\sigma_{W,L:n,m}$) whenever $n$ and $m$ are clear from the context.

## 4 Main Results

We first present our main approximation theoretic guarantee, which gives complexity estimates for mixtures of MLPs with trainable PReLU activation functions when uniformly approximating arbitrary locally-Hölder function on the closed unit ball of $\mathbb{R}^n$, defined by $\overline{B}_n(0,1) \stackrel{\text{def.}}{=} \{x \in \mathbb{R}^n : \|x\| \leq 1\}$.

Our rates depend on a "number of experts-to-expert complexity trade-off parameter" $r \in \mathbb{R}$ which determines how fast the overall MoE complexity scales, in terms of the number of experts and the complexity of each expert, as the approximation error becomes small. Setting $r < 0$ implies that more model complexity will be loaded into each expert MLP and there will be fewer experts defining the MoE. In contrast, setting $r > 0$ loads less complexity in each expert MLP at the cost of more experts in the MoE. In particular, when $r = 0$, each expert will have constant complexity even when the approximation error becomes arbitrarily small.

**Algorithm 1:** Routing Tree from Example 1.

**1** **if** $q_{1,1} = 1$ **then**
**2**   **if** $q_{2,1} = 1$ **then**
**3**     Index $\leftarrow 1$
**4**   **else**
**5**     Index $\leftarrow 2$
**6**   **end if**
**7** **else**
**8**   **if** $q_{2,2} = 1$ **then**
**9**     Index $\leftarrow 2$
**10**   **else**
**11**     Index $\leftarrow 3$
**12**   **end if**
**13** **end if**

**Theorem 4.1** (Trade-Off: No. Expert vs. Expert Complexity)**.** *Suppose that $\sigma$ satisfies Definition 3.1. Fix an "number of experts-to-expert complexity trade-off parameter" $r \in \mathbb{R}$. For every $\alpha$-Hölder map $f : \overline{B}_n(0, 1) \to \mathbb{R}^m$ with $0 < \alpha \leq 1$ and each approximation error $\varepsilon > 0$, there is a $p \in \mathbb{N}_+$, a binary tree $\mathcal{T} \stackrel{\text{def.}}{=} (V, E)$ with leaves $\mathcal{L} \stackrel{\text{def.}}{=} \{(v_i, \theta_i)\}_{i=1}^L \subseteq \overline{B}_n(0, 1) \times \mathbb{R}^p$ and a family of MLPs with (P)ReLU activation function $\{\hat{f}_{\theta_i}\}_{i=1}^L$ defined by $p$ parameters and mapping $\mathbb{R}^n$ to $\mathbb{R}^m$ satisfying:*

$$\max_{x \in \overline{B}_n(0,1)} \min_{(v_i, \theta_i) \in \mathcal{L}} \|x - v_i\| \in \Theta\big(\varepsilon^{1/\alpha - 2r/n}\big)$$

*and for each $x \in \overline{B}_n(0, 1)$ and $i = 1, \ldots, L$, if $\|x - v_i\| < \delta$ then*

$$\|f(x) - f_{\theta_i}(x)\| < \varepsilon.$$

*The depth and width of each $\hat{f}_{\theta_i}$ and the number of leaves, height, and number of nodes required to build the binary tree are all recorded in Table 1.*

A more general version of Theorem 4.1 is presented below as Theorem 5.3. In this version of our main approximation theorem, the target function can be any arbitrary continuous function defined on a non-empty compact subset of $\mathbb{R}^n$, and the routing tree can be $\nu$-ary for any natural number $\nu \geq 2$.

Next, we demonstrate that the MoMLP model can generalize and generate functions that are PAC-learnable, thanks to its finite VC dimension. This property, however, breaks down in MLP models with super-expressive activation functions.

**Theorem 4.2** (VC-Dimension Bounds for MoMLPs - MoMLPs Can Generalize)**.** *Let $J, W, L, n \in \mathbb{N}_+$. Then* VC $\big(\mathcal{NN}_{J,W,L:n,1}^{\text{PReLU}}\big)$ *is of*

$$\mathcal{O}\big(L \log(L)^2 \max\{nL \log(L), JW^2 \log(JW)\}\big) \tag{8}$$

*In particular,* VC$(\mathcal{NP}_{J,W,L:n,1}^{ReLU}) < \infty$.

## 4.1 Discussion

**Trade-off between Number of Experts and Expert Complexity.** Our results suggest that, theoretically, successful MoE models may not need each expert to be highly overparameterized if there are enough experts. This hypothesis is ablated experimentally in Section 6 in the context of irregular function approximation in low-dimension space; which is equivalent to high-dimensional regular function approximation (see Appendix C for a discussion on this later point).

**Pruning.** Additionally, one might consider the option of pruning a sizable model, conceivably trained on a GPU with a larger VRAM, for utilization on a smaller GPU during inference as an alternative to our method. Nevertheless, in frameworks like PyTorch, pruning does not result in a reduction of operations, acceleration, or diminished VRAM memory usage. Instead, pruning only masks the original model weights with zeros. The reduction in model size occurs only when saved in offline memory in sparse mode, which, in any case, is not a significant concern.

**Logarithmic number of queries via trees.** For many prototypes, as in our main guarantee, the MoMLPs only need to evaluate the distance between the given input and a logarithmic number of prototypes—specifically, one for each level in the tree—when using deep binary trees to hierarchically refine the Voronoi cells. Thus, a given machine never processes the exponential number of prototypes, and only $\nu \lceil \log_\nu(K) \rceil$ prototypes are ever queried for any given input; when trees are $\nu$-ary (as in Theorem 5.3), and where $K$ denotes the number of prototypes. Since we consider that prototypes are queried separately and before loading MoMLPs, we do not take them into account when counting the number of learnable parameters. Moreover, the size of our prototypes is negligible in our experiments.

**Functions which can be approximated by MoEs with small numbers of experts.** Our theoretical analysis adopts a worst-case perspective, focusing on the most difficult-to-approximate function within a given $\alpha$-Hölder class. Consequently, the number of experts in the MoE can become very large. However, this is often not the case in practice or in our experiments presented in Section 6. We anticipate that favorable

*Table 2:* VC Dimension of the MoMLPs, ReLU MLP, and MLP model with Super-Expressive Activation function of Shen et al. (2022a). All models have depth $J$, width $W$, and (when applicable) $L$ leaves; where $J, W, L, n \in \mathbb{N}_+$.

| Model | VC Dim | Ref. |
|---|---|---|
| MoMLPs | $\mathcal{O}\big(L\log(L)^2\,\max\{nL\log(L), JW^2\log(JW)\}\big)$ | Thrm 4.2 |
| ReLU MLP | $\mathcal{O}\big(JW^2\log(JW)\big)$ | Bartlett et al. (2019) |
| Super-Expressive | $\infty$ | Prop 4.4 |

approximation guarantees can be achieved with only a small number of expert ReLU MLPs, each having a limited number of non-zero parameters. Similar to the approximation theory for ReLU MLPs, we expect this to hold when the target function is sufficiently smooth, such as functions belonging to certain Besov spaces (Suzuki, 2018; Gühring & Raslan, 2021; Siegel & Xu, 2022), or when the function's structure aligns well with that of deep neural networks (Mhaskar et al., 2017; Cheridito et al., 2021b). Exploring these scenarios represents an interesting research direction that complements the worst-case analysis presented in this manuscript.

### 4.2 Application: Controlling The Complexity in VRAM maintaining a Finite VC Dimension

**Super-Expressive Activation Functions Have Infinite VC-Dimension.** We complement the main result of Bartlett et al. (2019) by demonstrating that the class of unstable MLPs (Shen et al., 2022a) possesses infinite VC dimension, even when they have finite depth and width. Thus, while they may serve as a gold standard from the perspective of approximation theory, they should not be considered a benchmark gold standard from the viewpoint of learning theory.

We consider a mild extension of the super-expressive activation function of Shen et al. (2022a). This parametric extension allows it to implement the identity map on the real line as well as the original super-expressive activation function thereof.

**Definition 4.3** (Trainable Super-Expressive Activation Function)**.** A trainable action function $\sigma : \mathbb{R} \times \mathbb{R} \to \mathbb{R}$ is of super-expressive type if for all $\gamma \in \mathbb{R}$

$$\sigma_\gamma : \mathbb{R} \ni x \mapsto \gamma x + (1-\gamma)\sigma^\star(x)$$

where $\sigma^\star : \mathbb{R} \to \mathbb{R}$ is given by

$$\sigma^\star(x) \stackrel{\text{def.}}{=} |x \bmod(2)| I_{x \in [0,\infty)} + \frac{x}{|x|+1} I_{x \in (-\infty, 0)} \tag{9}$$

**Proposition 4.4** (MLPs with Super-Expressive Activation Do Not Generalize)**.** *Let $\mathcal{F}$ be the set of MLPs with activation function in Definition 4.3, depth at-most 15, and width at-most $36n(2n+2)$. Then $\mathrm{VC}(\mathcal{F}) = \infty$.*

The VC dimension bounds for the standard MLP model, MLP with a super-expressive activation function as proposed by Shen et al. (2022a), and the MoMLP model are summarized in Table 2.

## 5 Overview of Derivation

We now overview the proof of our main result and its full technical formulation. These objectives require us to recall definitions from the analysis of metric spaces, which were not required in the statement of our main result but which are required when overviewing our proof.

### 5.1 Technical Definitions

The metric ball in $(\mathcal{X}, d)$ of radius $r > 0$ at $x \in \mathcal{X}$ is denoted by $\mathrm{Ball}_{(\mathcal{X},d)}(x, r) \stackrel{\text{def.}}{=} \{z \in \mathcal{X} : d(x, z) < r\}$. A metric space $(\mathcal{X}, d)$ is called *doubling*, if there is $C \in \mathbb{N}_+$ for which every metric ball in $(\mathcal{X}, d)$ can be covered

by at most $C$ metric balls of half its radius. The smallest such constant is called $(\mathcal{X}, d)$'s *doubling number*, and is here denoted by $C_{(\mathcal{X},d)}$. Though this definition may seem abstract at first, Heinonen (2001, Theorem 12.1) provides an almost familiar characterization of all doubling metric spaces; indeed, $\mathcal{K}$ is a doubling metric space if and only if it can be identified via a suitable invertible map[1] with a subset of some Euclidean space. Every subset of $\mathbb{R}^n$, for any $n \in \mathbb{N}_+$, is a doubling metric space; see Robinson (2011, Chapter 9) for details.

*Example* 2 (Subsets of Euclidean Spaces). Fix a dimension $n \in \mathbb{N}_+$. The doubling number of any subset of Euclidean space is[2] at most $2^{n+1}$.

In what follows, all *logarithms* will be taken *base* 2, unless explicitly stated otherwise, i.e. $\log_v$ is base $v$ for a given $v \in \mathbb{N}_+$ and $\log \overset{\text{def.}}{=} \log_2$. As in Petrova & Wojtaszczyk (2023, page 762), the radius of a subset $A \subseteq \mathbb{R}^n$, denoted by $\mathrm{rad}(A)$, is defined by

$$\mathrm{rad}(A) \overset{\text{def.}}{=} \inf_{x \in \mathbb{R}^n} \sup_{a \in A} \|x - a\|. \tag{10}$$

The diameter of any such set $A$, denoted by $\mathrm{diam}(A)$, satisfies the inequality $\mathrm{diam}(A) \leq 2\,\mathrm{rad}(A)$.

Finally, let us recall the notion of a uniformly continuous function. Fix $n, m \in \mathbb{N}_+$ and let $X \subset \mathbb{R}^n$. Let $\omega : [0, \infty) \to [0, \infty)$ be a monotonically increasing function which is continuous at 0 and satisfies $\omega(0) = 0$. Such an $\omega$ is called a *modulus of continuity*. A function $f : X \to \mathbb{R}^m$ is said to be $\omega$-uniformly continuous if

$$\|f(x) - f(y)\| \leq \omega\big(\|x - y\|\big)$$

holds for all $x, y \in \mathcal{X}$. We note that every continuous function is uniformly continuous if $\mathcal{X}$ is compact and that its modulus of continuity may depend on $\mathcal{X}$. Furthermore, we note that every $(\alpha, L)$-Hölder function is uniformly continuous with modulus of continuity $\omega(t) = L\,t^\alpha$.

## 5.2 Helping to Explain MoEs via Proof Sketch

**Lemma 5.1** (Size of a Tree Whose Nodes Form a $\delta$-net of a Compact Subset of $\mathbb{R}^n$). *Let $\mathcal{K}$ be a compact subset of $\mathbb{R}^n$ whose doubling number is $C$. Fix $v \in \mathbb{N}$ with $v \geq 2$, and $0 < \delta \leq \mathrm{rad}(\mathcal{K})$. There exists an $v$-ary tree $\mathcal{T} \overset{\text{def.}}{=} (V, E)$ with leaves $\mathcal{L} \subseteq \mathcal{K}$ satisfying*

$$\max_{x \in \mathcal{K}} \min_{v \in \mathcal{L}} \|x - v\| < \delta. \tag{11}$$

*Furthermore, the number of leaves $L \overset{\text{def.}}{=} \#\mathcal{L}$, height, and total number of nodes $\#V$ of the tree $\mathcal{T}$ are*

*(i)* ***Leaves:*** *at most $L = v^{\left\lceil c\,\log(C)\left(1 + \log(\delta^{-1}\,\mathrm{diam}(\mathcal{K}))\right)\right\rceil}$,*

*(ii)* ***Height:*** *$\left\lceil c\,\log(C)\big(1 + \log(\delta^{-1}\,\mathrm{diam}(\mathcal{K}))\big)\right\rceil$,*

*(iii)* ***Nodes:*** *At most $\dfrac{v^{\left\lceil c\,\log(C)(1+\log(\delta^{-1}\,\mathrm{diam}(\mathcal{K})))\right\rceil + 1} - 1}{\left\lceil c\,\log(C)\left(1 + \log(\delta^{-1}\,\mathrm{diam}(\mathcal{K}))\right)\right\rceil - 1}$*

*where $c \overset{\text{def.}}{=} 1/\log(v)$.*

At each node of the tree, we will place an MLP which only locally approximates the target function on a little ball of suitable radius (implied by the tree valency $v$ and height $h$) of lemma 5.1. I.e. by the storage space we would like to allocate to our MoMLP model. The next step of the proof relies on a mild extension of the *quantitative universal approximation theorem* in Shen et al. (2022a); Lu et al. (2021a) to the multivariate case, as well as an extension of the multivariate approximation result of Acciaio et al. (2023, Proposition 3.10) beyond the Hölder case.

**Lemma 5.2** (Vector-Valued Universal Approximation Theorem with Explicit Diameter Dependence). *Let $n, m \in \mathbb{N}_+$ with $n \geq 3$, $\mathcal{K} \subseteq \mathbb{R}^n$ be compact set of radius $\delta \geq 0$, $f : \mathcal{K} \to \mathbb{R}^m$ be uniformly continuous with strictly monotone continuous modulus of continuity $\omega$. Let $\sigma$ be an activation function as in Definitions 4.3*

---

[1]So called quasi-symmetric maps, see Heinonen (2001, page 78).

[2]See Robinson (2011, Lemma 9.2) and the brief computations in the proof of Robinson (2011, Lemma 9.4).

*or 3.1. For each $\varepsilon > 0$, there exists an MLP $\hat{f}_\theta : \mathbb{R}^n \to \mathbb{R}^m$ with trainable activation function $\sigma$ satisfying the uniform estimate*

$$\sup_{x \in \mathcal{K}} \|f(x) - \hat{f}_\theta(x)\| \leq \epsilon.$$

*The depth and width of $\hat{f}$ are recorded in Table 3.*

Table 3: Complexity of the MLP $\hat{f}_\theta$ in Lemma 5.2. See Table 7 in Appendix A for more detailed estimates.

| Activation $\sigma$ | Super Expressive 4.3 | PReLU 3.1 |
|---|---|---|
| Depth $(J)$ | $\mathcal{O}(1)$ | $\mathcal{O}\big((\delta/\omega^{-1}(\varepsilon))^{n/2}\big)$ |
| Width $(\max_j d_j)$ | $\mathcal{O}(1)$ | $\mathcal{O}(1)$ |

Combining Lemmata 5.1 and 5.2 we obtain Theorem 4.1. We now present the technical version of Theorem 4.1. This result allows distributed neural computing using $\nu$-ary trees and allows for the approximation general uniformly continuous target functions.

**Theorem 5.3** (Trade-Off: No. Expert vs. Expert Complexity - Technical Version of Theorem 4.1)**.** *Suppose that $\sigma$ satisfies Definition 3.1. Let $\mathcal{K}$ be a compact subset of $\mathbb{R}^n$ whose doubling number is $C$. Fix an "number of experts-to-expert complexity trade-off parameter" $r \in \mathbb{R}$ and a "valency parameter" $\nu \in \mathbb{N}$ with $\nu \geq 2$.*
*For every uniformly continuous map $f : \mathcal{K} \to \mathbb{R}^m$ with modulus of continuity $\omega$ and each approximation error $\varepsilon > 0$, $p \in \mathbb{N}_+$, there is an $\nu$-ary tree $\mathcal{T} \stackrel{\text{def.}}{=} (V, E)$ with leaves $\mathcal{L} \stackrel{\text{def.}}{=} \{(v_i, \theta_i)\}_{i=1}^L \subseteq \mathcal{K} \times \mathbb{R}^p$ and a family of MLPs with (P)ReLU activation function $\{\hat{f}_{\theta_i}\}_{i=1}^L$ defined by $p$ parameters mapping $\mathbb{R}^n$ to $\mathbb{R}^m$ satisfying:*

$$\max_{x \in \mathcal{K}} \min_{(v_i, \theta_i) \in \mathcal{L}} \|x - v_i\| < \frac{\varepsilon^{-2r/n}}{2} \, \omega^{-1}\left(\frac{\varepsilon}{131 \, (nm)^{1/2}}\right)$$

*and for each $x \in \mathcal{K}$ and $i = 1, \ldots, L$, if $\|x - v_i\| < \delta$ then*

$$\|f(x) - f_{\theta_i}(x)\| < \varepsilon.$$

*The depth and width of each $\hat{f}_{\theta_i}$ and the number of leaves, height, and number of nodes required to build the $\nu$-ary tree are all recorded in Table 1.*

# 6 Does one Need Overparameterized Experts if there are Enough Experts?

We evaluate our approach in two standard machine learning tasks: regression and classification. We experimentally show that MoMLPs, which distribute predictions over multiple neural networks, are competitive with a single large neural network containing as many model parameters as all the MoMLPs combined. This is desirable in cases where the large neural network does not fit into the memory of a single machine. On the contrary, the MoMLP model can be trained by distributing each MoMLP on separate machines (or equivalently serially on a single machine). Inference can then be performed by loading only a single MoMLP at a time into the GPU.

## 6.1 Regression

We first consider regression, where the goal is to approximate non-convex synthetic functions often used for performance test problems. In particular, we choose 1-dimensional Hölder functions, as well as Ackley (Ackley, 1987) and Rastrigin (Rastrigin, 1974) functions, whose formulations are detailed in Appendix D.1.

**1D Hölder Functions.** We illustrate our primary finding, as encapsulated in Theorem 4.1, by leveraging 1-dimensional functions characterized by very low regularity. This choice is motivated by the jagged structure inherent in such functions, necessitating an exponentially higher sampling frequency compared to smooth

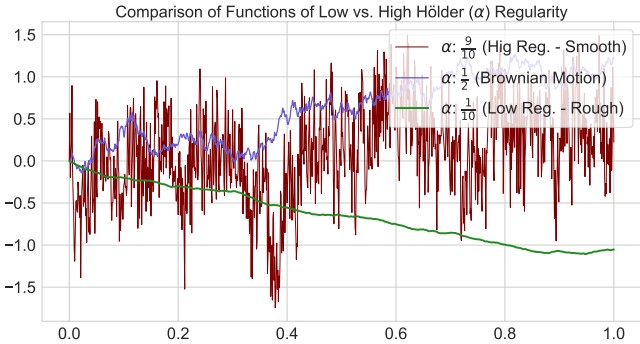

*Figure 2:* Visual Comparison of Functions with High ($\alpha \approx 1$) vs. Low ($\alpha \approx 0$) Hölder regularity. If $\alpha \approx 1$, the function (green) is approximately differentiable almost everywhere, meaning it does not osculate much locally and thus is simple to approximate. If $\alpha \approx 0$, the function may be nowhere differentiable and jagged; its extreme details make it difficult to approximate.

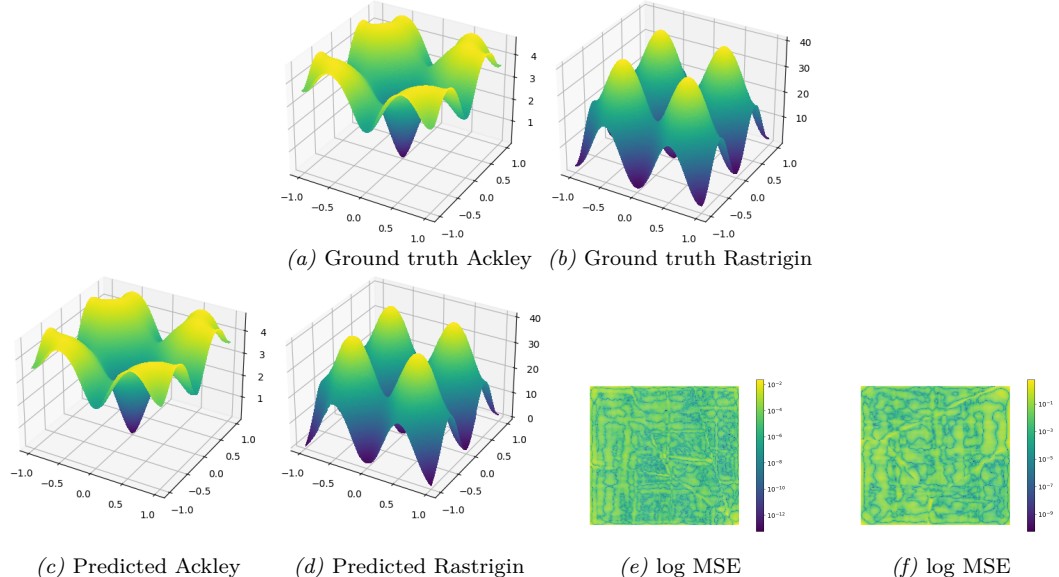

*(a)* Ground truth Ackley    *(b)* Ground truth Rastrigin

*(c)* Predicted Ackley    *(d)* Predicted Rastrigin    *(e)* log MSE    *(f)* log MSE

*Figure 3:* Comparison of ground truth and predicted results for 2D Ackley and Rastrigin functions over the domain $[-1, 1]^2$.

functions for achieving an accurate reconstruction. Indeed, this crucial sampling step forms the foundation of many quantitative universal approximation theorems (Yarotsky, 2017; Shen et al., 2021; Kratsios & Papon, 2022). As elaborated in Appendix C, approximating a well-behaved (Lipschitz) function in $d$ dimensions poses a challenge equivalent to approximating a highly irregular function ($1/d$-Hölder) in a single dimension. A visual representation of a $1/d$-Hölder function is presented in Figure 2, exemplified by the trajectory of a *fractional Brownian motion* with a Hurst parameter of $\alpha = 1/d$. A formal definition is available in Appendix C.

**2D and 3D Functions.** We select the Ackley and Rastrigin functions, with their respective 2D representations showcased in Figure 3, as widely recognized benchmarks in the field of optimization.

**Evaluation protocol.** We consider the setup where the domain of a function that we try to approximate is the $n$-dimensional closed set $[a, b]^n$. For instance, we arbitrarily choose the domain $[0, 1]$ when $n = 1$, and $[-1, 1]^n$ when $n \geq 2$. Our training and test samples are the $s^n$ vertices of the regular grid defined on $[a, b]^n$. At each run, 80% of the samples are randomly selected for training and validation, and the remaining 20% for testing. During training, for a given and fixed set of $K$ prototypes $p \overset{\text{def.}}{=} (p_1, \ldots, p_K)$, we assign each training

sample $x$ to its nearest prototype $p_k$ and associated neural network $\hat{f}_k$. We learn the prototypes as explained in Appendix D.2. For simplicity, we set the number of prototypes to $K = 4$; we also set $s = 10,000$ if $n = 1$, $s = 150$ if $n = 2$ (i.e., $150^2 = 22,500$ samples), and $s = 30$ if $n = 3$ (i.e., $27,000$ samples). More details can be found in Appendix D.

**Test performance.** In Table 4, we present the mean squared error obtained on the test set across 10 random initializations and various splits of the training/test sets. The baseline consists of a single neural network with the same overall architecture as each MoMLP but possesses as many parameters as all the MoMLPs combined. In all instances, the MoMLP model demonstrates a significant performance advantage compared to the baseline. Figure 3 illustrates the predictions generated by our MoMLPs, showcasing the capability of our approach to achieve a good approximation of the ground truth functions.

## 6.2   Classification

**Datasets.** We evaluate classification on standard image datasets such as CIFAR-10 (Krizhevsky & Hinton, 2010), CIFAR-100, and Food-101 (Bossard et al., 2014), which consist of 10, 100, and 101 different classes, respectively. We use the standard splits of training/test sets: the datasets include (per category) 5,000 training and 1,000 test images for CIFAR-10, 500 training and 100 test images for CIFAR-100, and 750 and 250 for Food-101.

**Training.** Our MoMLP model takes as input latent DINOv2 encodings (Oquab et al., 2023) of images from the aforementioned datasets. Each sample $x \in \mathbb{R}^{768}$ corresponds to a DINOv2 embedding (i.e., $n = 768$). Additionally, we set the prototypes as centroids obtained through the standard $K$-means clustering on the DINOv2 embedding space. The replacement of our original prototype learning algorithm is sensible in this context, as we operate within a structured latent space optimized through self-supervised learning using large-scale compute and datasets.

Due to the potential class imbalance in the various Voronoi cells formed by the prototypes, we utilize two variations of the cross-entropy loss for each MoMLP $\hat{f}_k$. The first variation, termed *unweighted*, assigns equal weight to all categories. The *weighted* variation assigns a weight that is inversely proportional to the distribution of each category in the Voronoi cell defined by the prototype $p_k$.

**Test performance.** We present the test classification accuracy over 10 different runs (with random initialization) of both the baseline and our MoMLPs in Table 5. The weighted version performs slightly better on datasets with a large number of categories. Nonetheless, our approach achieves comparable results with the baseline (i.e., no difference with statistical significance), effectively decomposing the prediction across multiple smaller models while requiring less VRAM per neural network. We report the same type of experiment on the CIFAR datasets at the pixel level in Appendix D.4.

## 6.3   Discussion

Our experiments in the above subsections demonstrate that MoMLPs can outperform or match the performance of a single large neural network with the same overall architecture and an equivalent total number of parameters, which aligns with our theoretical insights. This advantage is particularly significant in scenarios where a single large neural network cannot be stored on a given machine or cluster due to its high VRAM requirements, a common challenge in the context of large language models and other recent large-scale models. While this work focuses on smaller-scale experiments, we believe our theoretical framework represents an important initial step toward addressing and understanding these challenges from a mathematically principled perspective.

*Table 4:* Test mean squared error (average and standard deviation) for the different functions of the regression task.

|  | 1D Hölder | 2D Ackley | 3D Ackley | 2D Rastrigin | 3D Rastrigin |
|---|---|---|---|---|---|
| MoMLPs (ours) | **$0.057 \pm 0.085$** | **$0.00015 \pm 0.00006$** | **$0.00068 \pm 0.00010$** | **$0.0480 \pm 0.0073$** | **$1.0062 \pm 0.0446$** |
| Baseline | $0.128 \pm 0.012$ | $0.08723 \pm 0.01059$ | $0.09303 \pm 0.03156$ | $3.0511 \pm 0.3581$ | $8.0376 \pm 4.0499$ |

*Table 5:* Test classification accuracy using DINOv2 features as input (average and standard deviation).

| Dataset | CIFAR-10 | CIFAR-100 | Food-101 |
|---|---|---|---|
| Ours (weighted) | $98.40 \pm 0.05$ | $90.01 \pm 0.11$ | $91.86 \pm 0.10$ |
| Ours (unweighted) | $98.42 \pm 0.04$ | $89.62 \pm 0.25$ | $91.79 \pm 0.16$ |
| Baseline | $98.45 \pm 0.06$ | $89.85 \pm 0.17$ | $91.45 \pm 1.09$ |

The concept of decomposing a single large neural network into multiple smaller models that run in parallel has been successfully applied in domains such as computer vision, as demonstrated in works like (Ren et al., 2024; Song et al., 2024). However, existing studies in the literature are largely empirical and application-focused, lacking the theoretical approximation rates provided by our work.

## 7 Conclusion

We presented approximation-theoretic and statistical foundations for MoEs by analysing the MoMLP model. We found that MoMLPs can achieve arbitrary uniform approximation accuracy of continuous functions on compact subsets of Euclidean space while maintaining a feasible number of parameters in active VRAM memory (Theorem 5.3). However, this naturally comes at the cost of requiring an exponential number of experts. We obtain upper bounds on the VC dimension of the MoMLP model (Theorem 4.2), akin to the results of Bartlett et al. (2019) for ReLU MLPs, showing that deep-learning-based MoEs can generalize.

## 8 Acknowledgment and Funding

The authors thank Gabriel Conant, James Lucas, and Rafid Mahmood for their helpful feedback while preparing the manuscript.

AK acknowledges financial support from an NSERC Discovery Grant No. RGPIN-2023-04482 and their McMaster Startup Funds. AK also acknowledges that resources used in preparing this research were provided, in part, by the Province of Ontario, the Government of Canada through CIFAR, and companies sponsoring the Vector Institute https://vectorinstitute.ai/partnerships/current-partners/. HSOB acknowledges financial support from an EV travel grant.

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

# A    Detailed Tables And Rates

*Table 6:* Complexity of Feedforward Neural Network $\hat{f}_\theta$ and the $\nu$-ary routing tree in Theorem 5.3. Here $c \stackrel{\text{def.}}{=} \log(v)^{-1}$. See Table 8 in Appendix A for more detailed estimates.

| Parameter | Estimate |
|---|---|
| Depth ($J$) | $\mathcal{O}(\max\{1, \varepsilon^{-r}\})$ |
| Width ($\max_j d_j$) | $\mathcal{O}(1)$ |
| No. Experts | $\mathcal{O}\big(\max\{1, \varepsilon^{2r/n}/\omega^{-1}(\varepsilon)\}\big)$ |
| Routing Complexity | $\mathcal{O}\big(\max\{1, \log(\varepsilon^{2r/n}/\omega^{-1}(\varepsilon))\}\big)$ |

*Table 7:* Complexity of the MLP $\hat{f}_\theta$ in Lemma 5.2.

| Activation $\sigma$ | Super Expressive 4.3 | PReLU 3.1 |
|---|---|---|
| Depth ($J$) | $15m$ | $m\left(19 + 2n + 11\left\lceil\left(\frac{\delta\,2^{3/2}n^{1/2}}{(n+1)^{1/2}\omega^{-1}\big(\varepsilon/(131\sqrt{n\,m})\big)}\right)^{n/2}\right\rceil\right)$ |
| Width ($\max_j d_j$) | $36n(2n+1)+m$ | $16\max\{n,3\}+m$ |

*Table 8:* Complexity of Feedforward Neural Network $\hat{f}_\theta$ and the $\nu$-ary tree in Theorem 5.3. Here $c \stackrel{\text{def.}}{=} \log(v)^{-1}$.

| Parameter | Estimate |
|---|---|
| Depth ($J$) | $m\left(19 + 2n + 11\lceil \varepsilon^{-r}\rceil\right)$ |
| Width ($\max_j d_j$) | $16\max\{n,3\}+m$ |
| No. Experts (No. Leaves) | $\mathcal{O}\Big(v^{\big\lceil c\,\log(C)\big(1+\log(\varepsilon^{2r/n}\,\text{diam}(\mathcal{K}))/\big(2\,\omega^{-1}\big(\frac{\varepsilon}{131\,(nm)^{1/2}}\big)\big)\big)\big\rceil},\Big)$ |
| Height (Routing Complexity) | $\left\lceil c\,\log(C)\big(1 + \log(\epsilon^{2r/n}\,\text{diam}(\mathcal{K})/\big(2\omega^{-1}\big(\frac{\varepsilon}{131\,(nm)^{1/2}}\big)\big)\big)\right\rceil$ |
| Nodes | $\mathcal{O}\left(\frac{v^{\lceil c\,\log(C)(1+\log(\varepsilon^{2r/n}\,\text{diam}(\mathcal{K})/\big(2\omega^{-1}(\frac{\varepsilon}{131\,(nm)^{1/2}})\big)))\rceil+1}-1}{\left\lceil c\log(C)\big(1+\log(\frac{\varepsilon^{2r/n}}{2}\,\text{diam}(\mathcal{K}))/\omega^{-1}\big(\frac{\varepsilon}{131\,(nm)^{1/2}}\big)\big)\right\rceil-1}\right)$ |

# B    Appendix: Proofs

## B.1    Proof of Theorem 4.1

*Proof of Lemma 5.1.* Since $\mathcal{K}$ is a doubling metric space then, (Acciaio et al., 2023, Lemma 7.1), for each $\delta > 0$, there exist $x_1, \ldots, x_N \in \mathcal{K}$ satisfying

$$\max_{x\in\mathcal{K}}\ \min_{i=1,\ldots,N_\delta}\ \|x - x_i\| < \delta \text{ and } N_\delta \leq C^{\lceil\log(\text{diam}(\mathcal{K})/\delta)\rceil}.$$

In particular, since the doubling number of $\mathcal{K}$ is $C$, we have the upper-bound

$$N_\delta \leq C\,C^{\log(\delta^{-1}\,\text{diam}(\mathcal{K}))}. \tag{12}$$

An elementary computation shows that the complete $v$-ary tree of height $h$ has leaves $L$ and total vertices/nodes $V$ given by

$$L = v^h \text{ and } V = \frac{v^{h+1}-1}{h-1}. \tag{13}$$

Taking the formulation of $L$ given in equation 13, to be the least *integer* upper bound of the right-hand side of equation 12, which is itself an upper-bound for $N_\delta$, and solving for $h$ yields:

$$h = \left\lceil \log_v(C)\big(1 + \log(\delta^{-1} \operatorname{diam}(\mathcal{K}))\big) \right\rceil, \tag{14}$$

where the integer ceiling was applied since $h$ must be an integer.

Let $\mathcal{L}$ be any set $v^h$ points in $\mathcal{K}$ containing the set $\{x_i\}_{i=1}^{N_\delta}$. Let $\mathcal{T} \stackrel{\text{def.}}{=} (V, E)$ be any complete binary tree with leaves $\mathcal{L}$; note that, $\mathcal{L} \subseteq V$. By construction, and the computation in equation 13, $\mathcal{T}$ has $v^h$ leaves and $\frac{Lv-1}{h-1}$ nodes. $\qquad\square$

For completeness, we include a minor modification of the proof of Acciaio et al. (2023, Proposition 3.10), which allows for the approximation of uniformly continuous functions of arbitrarily low regularity. The original formulation of that result only allows for $\alpha$-Hölder function.

*Proof of Lemma 5.2.* If $f(x) = c$ for some constant $c > 0$, then the statement holds with the neural network $\hat{f}(x) = c$, which can be represented as in equation 2 with $[d] = (n, m)$, where $A^j$ is the 0 matrix for all $j$, and the "$c$" in equation 2 is taken to be this constant $c$. Therefore, we henceforth only need to consider the case where $f$ is not constant. Let us observe that, if we pick some $x^\star \in \mathcal{K}$, then for any multi-index $[d]$ and any neural network $\hat{f}_\theta \in \mathcal{NN}_{[d]}^\sigma$, $\hat{f}_\theta(x) - f(x^\star) \in \mathcal{NN}_{[d]}^\sigma$, since $\mathcal{NN}_{[d]}^\sigma$ is invariant to post-composition by affine functions. Thus, we represent $\hat{f}_\theta(x) - f(x^\star) = \hat{f}_{\theta^\star}(x)$, for some $\theta^\star \in \mathbb{R}^{P([d])}$. Consequently:

$$\sup_{x \in \mathcal{K}} \left| \|(f(x) - f(x^\star)) - \hat{f}_{\theta^\star}(x)\| - \|f(x) - \hat{f}_\theta(x)\| \right| = 0.$$

Therefore, without loss of generality, we assume that $f(x^*) = 0$ for some $x^* \in \mathcal{K}$. By Benyamini & Lindenstrauss (2000, Theorem 1.12), there exists an $\omega$-uniformly continuous map $F : \mathbb{R}^n \to \mathbb{R}^m$ extending $f$.

**Step 1 – Normalizing $\tilde{f}$ to the Unit Cube:** First, we identify a hypercube "nestling" $\mathcal{K}$. To this end, let

$$r_\mathcal{K} \stackrel{\text{def.}}{=} \operatorname{diam}(\mathcal{K})\sqrt{\frac{n}{2(n+1)}}. \tag{15}$$

By Jung's Theorem (see Jung (1901)), there exists $x_0 \in \mathbb{R}^n$ such that the closed Euclidean ball $\overline{\operatorname{Ball}_{(\mathbb{R}^n, d_n)}(x_0, r_\mathcal{K})}$ contains $\mathcal{K}$. Therefore, by Hölder's inequality, we have that the $n$-dimensional hypercube $[x_0 - r_\mathcal{K}\bar{1}, x_0 + r_\mathcal{K}\bar{1}]$ [3]contains $\overline{B_{(\mathbb{R}^n, d_n)}(x_0, r_\mathcal{K})}$, where $\bar{1} = (1, \ldots, 1) \in \mathbb{R}^n$. Consequently, $\mathcal{K} \subseteq [x_0 - r_\mathcal{K}\bar{1}, x_0 + r_\mathcal{K}\bar{1}]$. Let $\tilde{f} \stackrel{\text{def.}}{=} F|_{[x_0 - r_\mathcal{K}\bar{1}, x_0 + r_\mathcal{K}\bar{1}]}$, then $\tilde{f} \in C([x_0 - r_\mathcal{K}\bar{1}, x_0 + r_\mathcal{K}\bar{1}], \mathbb{R}^m)$ is an $\omega$-uniformly continuous extension of $f$ to $[x_0 - r_\mathcal{K}\bar{1}, x_0 + r_\mathcal{K}\bar{1}]$.

Since $\mathcal{K}$ has at least two distinct points, then $r_\mathcal{K} > 0$. Hence, the affine function

$$T : \mathbb{R}^n \ni x \mapsto (2r_\mathcal{K})^{-1}(x - x_0 + r_\mathcal{K}\bar{1}) \in \mathbb{R}^n$$

is well-defined, invertible, not identically 0, and maps $[x_0 - r_\mathcal{K}\bar{1}, x_0 - r_\mathcal{K}\bar{1}]$ to $[0, 1]^n$. A direct computation shows that $g \stackrel{\text{def.}}{=} \tilde{f} \circ T^{-1}$ is also uniformly continuous, whose modulus of continuity $\tilde{\omega} : [0, \infty) \to [0, \infty)$ is given by

$$\tilde{\omega}(t) \stackrel{\text{def.}}{=} \omega(2r_\mathcal{K} t) \tag{16}$$

for all $t \in [0, \infty)$. Furthermore, since for each $i = 1, \ldots, m$, define $\operatorname{pj}_i : \mathbb{R}^m \ni y \mapsto y_i \in \mathbb{R}$. Since each $\operatorname{pj}_i$ is 1-Lipschitz then, for each $i = 1, \ldots, m$, the map $g_i \stackrel{\text{def.}}{=} \operatorname{pj}_i \circ g : [0, 1]^n \to \mathbb{R}$ is also $\tilde{\omega}$-uniformly continuous. By orthogonality, we also note that $g(x) = \sum_{i=1}^m g_i(x)\, e_i$, for each $x \in \mathbb{R}^n$, where $e_1, \ldots, e_m$ is the standard orthonormal basis of $\mathbb{R}^m$; i.e. the $i^{th}$ coordinate of $e_j$ is 1 if and only if $i = j$ and 0 is otherwise.

**Step 2 – Constructing the Approximator:** For $i = 1, \ldots, m$, let $\hat{f}_{\theta^{(i)}} \in \mathcal{NN}_{[d^{(i)}]}^\sigma$ for some multi-index

---

[3]For $x, y \in \mathbb{R}^n$ we denote by $[x, y]$ the hypercube defined by $\prod_{i=1}^n [x_i, y_i]$.

$[d^{(i)}] = (d_0^{(i)}, \ldots, d_J^{(i)})$ with $n$-dimensional input layer and 1-dimensional output layer, i.e. $d_0^{(i)} = n$ and $d_J^{(i)} = 1$, and let $\theta^{(i)} \in \mathbb{R}^{P([d^{(i)}])}$ be the parameters defining $\hat{f}_{\theta^{(i)}}$. Since the pre-composition by affine functions and the post-composition by linear functions of neural networks in $\mathcal{NN}_{[d^{(i)}]}^{\sigma}$ are again neural networks in $\mathcal{NN}_{[d^{(i)}]}^{\sigma}$, we have that $g_{\theta^{(i)}} \stackrel{\text{def.}}{=} \hat{f}_{\theta^{(i)}} \circ T^{-1}$ belongs to $\mathcal{NN}_{[d^{(i)}]}^{\sigma}$. Denote the standard basis of $\mathbb{R}^m$ by $\{e_i\}_{i=1}^m$. We compute:

$$
\begin{aligned}
&\sup_{x \in \mathcal{K}} \left\| f(x) - \sum_{i=1}^{m} \hat{f}_{\theta^{(i)}}(x) e_i \right\| \\
&= \sup_{x \in \mathcal{K}} \left\| \tilde{f}(x) - \sum_{i=1}^{m} \hat{f}_{\theta^{(i)}}(x) e_i \right\| \\
&\leq \sup_{x \in [x_0 - r_\mathcal{K} \bar{1}, x_0 + r_\mathcal{K} \bar{1}]} \left\| \tilde{f}(x) - \sum_{i=1}^{m} \hat{f}_{\theta^{(i)}}(x) e_i \right\| \\
&= \sup_{x \in [x_0 - r_\mathcal{K} \bar{1}, x_0 + r_\mathcal{K} \bar{1}]} \left\| \tilde{f} \circ T^{-1} \circ T(x) - \sum_{i=1}^{m} \hat{f}_{\theta^{(i)}} \circ T^{-1} \circ T(x) e_i \right\| \\
&= \sup_{u \in [0,1]^n} \left\| \sum_{i=1}^{m} g_i(u)\, e_i - \sum_{i=1}^{m} g_{\theta^{(i)}}(u) e_i \right\| \\
&\leq \sqrt{m} \max_{u \in [0,1]^n} \max_{1 \leq i \leq m} |g_i(u) - g_{\theta^{(i)}}(u)|.
\end{aligned}
\tag{17}
$$

Fix $\tilde{\varepsilon} > 0$, to be determined below. For each $i = 1, \ldots, m$, depending on which assumption $\sigma$ satisfies, (Shen et al., 2022b, Theorem 1.1) (resp. (Shen et al., 2022a, Theorem 1) if $\sigma$ is as in Definition 9) imply that there is a neural network with activation function $\sigma^\star : \mathbb{R} \to \mathbb{R}$ satisfying

$$
\max_{u \in [0,1]^n} |g_i(u) - g_{\theta^{(i)}}(u)| < \tilde{\varepsilon}.
\tag{18}
$$

Furthermore, the depth and width of these MLPs can be bounded above on a case-by-case basis as follows:

(i) If $\sigma$ satisfies Definition 4.3 then, setting each $\gamma = 0$ implies that $\sigma_0(x) = \sigma^\star(x)$, as defined in equation 9; thus

$$
J^{(i)} \leq 11 \text{ and } \max_{1 \leq j \leq J^{(i)}} d_j \leq 36n(2n+1)
$$

In this case, we set $\tilde{\varepsilon} \stackrel{\text{def.}}{=} \varepsilon/\sqrt{m}$; we have used Shen et al. (2022a, Theorem 1).

(ii) If $\sigma$ satisfies Definition 3.1 then, setting each $\gamma = 1$ implies that $\sigma_0(x) = \text{ReLU}(x) \stackrel{\text{def.}}{=} \max\{0, x\}$; yielding

$$
J^{(i)} \leq 18 + 2n + 11 \left\lceil \left( \frac{2r_\mathcal{K}}{\omega^{-1}\big(\varepsilon/(131\sqrt{n\,m})\big)} \right)^{n/2} \right\rceil \text{ and } \max_{1 \leq j \leq J^{(i)}} d_j \leq 16 \max\{n, 3\}
$$

in this case, we have employed Shen et al. (2022b, Theorem 1.1).

In either case, the estimate in equation 17 yields

$$
\max_{x \in \mathcal{K}} \left\| f(x) - \sum_{i=1}^{m} \hat{f}_{\theta^{(i)}}(x) e_i \right\| < \varepsilon.
$$

**Step 3 – Assembling into an MLP:** Let $g_1 \bullet g_2$ denote the component-wise composition of a univariate function $g_1$ with a multivariate function $g_2$.

If the activation function $\sigma$ is either in Definitions 4.3 or 3.1, then it trivially implements the identity $I_{\mathbb{R}}$ on $\mathbb{R}$ by setting $\gamma = 0$; i.e. $\sigma_1 = I_{\mathbb{R}}$. Consequentially, for any $k \in \mathbb{N}_+$, if $I_k$ denotes the $k \times k$-identity matrix, then $I_k \sigma_1 \bullet I_k \in \mathcal{NN}^\sigma_{[d_k]}$ with $P([d]) = 2k$, and $I_k \sigma_1 \bullet I_k = 1_{\mathbb{R}^k}$. Therefore, mutatis mutandis, $\mathcal{NN}^\sigma_{[\cdot]}$ satisfies the *c-identity requirement with*[4] $c = 2$, *as defined in* Cheridito et al. (2021a, Definition 4). From there, mutatis mutandis, we may apply Cheridito et al. (2021a, Proposition 5). Thus, there is a multi-index $[d] = (d_0, \ldots, d_J)$ with $d_0 = n$ and $d_J = m$, and a network $\hat{f}_\theta \in \mathcal{NN}^\sigma_{[d]}$ implementing $\sum_{i=1}^m \hat{f}_{\theta^{(i)}} e_i$, i.e.

$$\sum_{i=1}^m \hat{f}_{\theta^{(i)}} e_i = \hat{f}_\theta,$$

such that $\hat{f}_\theta$'s depth and width are bounded-above, on a case-by-case basis, by

(i) If $\sigma$ satisfies Definition 4.3 then, setting each $\gamma = 0$

$$J \leq 15\,m$$
$$\text{and} \quad \max_{1 \leq j \leq J^{(i)}} d_j \leq 36n(2n+1) + m.$$

In this case, we set $\tilde{\varepsilon} \stackrel{\text{def.}}{=} \varepsilon/\sqrt{m}$.

(ii) If $\sigma$ satisfies Definition 3.1 then, setting each $\gamma = 0$ yields

$$J \leq m\left(19 + 2n + 11\left\lceil\left(\frac{2r_{\mathcal{K}}}{\omega^{-1}\left(\varepsilon/(131\sqrt{n\,m})\right)}\right)^{n/2}\right\rceil\right)$$
$$\text{and} \quad \max_{1 \leq j \leq J^{(i)}} d_j \leq 16\,\max\{n, 3\} + m.$$

Incorporating the definition of $r_{\mathcal{K}}$ in equation 15 and employing the inequality $\text{diam}(\mathcal{K}) \leq 2\,\text{rad}(\mathcal{K})$ completes the proof. $\square$

**Lemma B.1** (Trade-Off: No. Expert vs. Expert Complexity). *Let $\mathcal{K}$ be a compact subset of $\mathbb{R}^n$ whose doubling number is $C$ and a uniformly continuous map $f : \mathcal{K} \to \mathbb{R}^m$ with modulus of continuity $\omega$.*

*Fix $v \in \mathbb{N}$ with $v \geq 2$, $0 < \delta \leq \text{rad}(\mathcal{K})$, and $\varepsilon > 0$. Suppose that $\sigma$ satisfies Definition 3.1. There exists a $p \in \mathbb{N}_+$ and a $v$-ary tree $\mathcal{T} \stackrel{\text{def.}}{=} (V, E)$ with leaves $\mathcal{L} \stackrel{\text{def.}}{=} \{(v_i, \theta_i)\}_{i=1}^L \subseteq \mathcal{K} \times \mathbb{R}^p$ satisfying*

$$\max_{x \in \mathcal{K}} \min_{(v_i, \theta_i) \in \mathcal{L}} \|x - v_i\| < \delta. \tag{19}$$

*Furthermore, for each $x \in \mathcal{K}$ and each $i = 1, \ldots, L$, if $\|x - v_i\| < \delta$ then*

$$\|f(x) - f_{\theta_i}(x)\| < \varepsilon.$$

*We have the following estimates:*

(i) **Depth.** *Depth of each $\hat{f}_{\theta_i}$ is $m\left(19 + 2n + 11\left\lceil\left(\frac{\delta 2^{3/2} n^{1/2}}{(n+1)^{1/2}\omega^{-1}\left(\varepsilon/(131\sqrt{n\,m})\right)}\right)^{n/2}\right\rceil\right)$*

(ii) **Width.** *Width of each $\hat{f}_{\theta_i}$ is $16\,\max\{n, 3\} + m$*

(iii) **Leaves:** *at most $L = v^{\left\lceil c\,\log(C)\left(1 + \log(\delta^{-1}\,\text{diam}(\mathcal{K}))\right)\right\rceil}$,*

---

[4]Formally, it satisfies what is the 1-identity requirement, thus it satisfies the *c*-identity requirement for all integers $c \geq 2$. However, the authors of Cheridito et al. (2021a) do not explicitly consider the extremal case where $c = 1$.

(iv) **Height:** $\lceil c \log(C)\big(1 + \log(\delta^{-1} \operatorname{diam}(\mathcal{K}))\big)\rceil$,

(v) **Nodes:** *At most* $\dfrac{v^{\lceil c \, \log(C)(1+\log(\delta^{-1} \operatorname{diam}(\mathcal{K})))\rceil + 1} - 1}{\left\lceil c \log(C)\Big(1 + \log(\delta^{-1} \operatorname{diam}(\mathcal{K}))\Big)\right\rceil - 1}$

*where* $c \overset{\text{def.}}{=} \log(v)^{-1}$.

*Proof of Lemma B.1.* Consider the tree $\tilde{\mathcal{T}}$ given by Lemma 5.1. For each leaf $v_i$ of $\tilde{\mathcal{T}}$, we apply Lemma 5.2 to deduce the existence of an MLP $\hat{f}_{\theta_i}$ with explicit depth and width estimates given by that lemma, satisfying the uniform estimate

$$\max_{\|x - v_i\| \leq \delta} \|f(x) - \hat{f}_{\theta_i}(x)\| < \varepsilon. \tag{20}$$

Let $\mathcal{T}$ be the same tree as $\tilde{\mathcal{T}}$ with leaves identified with $\{(v_i, \theta_i)\}_{i=1}^L$. $\qquad\square$

We now prove our main technical version, namely Theorem 5.3, which directly implies the special case recorded in Theorem 4.1.

*Proof of Theorem 5.3 (and this Theorem 4.1).* Applying Lemma B.1 with $\delta$ given as the solution of

$$\Big(\frac{\delta 2^{3/2} n^{1/2}}{(n+1)^{1/2} \omega^{-1}(\varepsilon/131 \, (nm)^{1/2})}\Big)^{n/2} \leq \Big(\frac{\delta 2^{3/2} n^{1/2}}{(2n^{1/2} \omega^{-1}(\varepsilon/131 \, (nm)^{1/2})}\Big)^{n/2} = \varepsilon^{-r}. \tag{21}$$

Solving equation 21 for $\delta$ implies that it is given by

$$\delta = \frac{\varepsilon^{-2r/n}}{2} \, \omega^{-1}\Big(\frac{\varepsilon}{131 \, (nm)^{1/2}}\Big).$$

This completes the proof. $\qquad\square$

*Proof of Theorem 4.1.* Setting $\mathcal{K} = B_n(0,1)$, $r = 1/2$, $\omega(t) = Lt$, and thus $\omega^{-1}(t) = L^{-1} t^{1/\alpha}$, in Theorem 5.3 yields the conclusion. Finally, by the computation in Example 2, we have that $C \leq 2^{n+1}$; thus, $\log(C) = (n+1)\log(2) \leq 2n$. Noting that $c = 1/\log(2) = 1$ completes the proof. $\qquad\square$

*Remark B.2.* The constant hidden under the big $\mathcal{O}$ in is $1 + \max\{1, \log(L^{1/\alpha} 262(nm)^{1/(2\alpha)})\}$.

## B.2 Proofs of Theorem 4.2

We use the following lemma and its proof due to *Gabriel Conant.*

**Lemma B.3** (Conant (2023)). *Fix* $n, L, d \in \mathbb{N}_+$. *Let* $\mathcal{H}$ *be a non-empty set of functions from* $\mathbb{R}^n$ *to* $\{0,1\}$ *of VC dimension at-most* $d$. *Let* $\mathcal{C}_L$ *be the set of all ordered partitions (Voronoi diagrams)* $(C_l)_{l=1}^{\tilde{L}}$ *covering* $\mathbb{R}^n$, *where* $\tilde{L} \leq L$, *and for which there exist distinct* $p_1, \dots, p_{\tilde{L}} \in \mathbb{R}^n$ *such that: for each* $l = 1, \dots, \tilde{L}$

$$\begin{aligned} C_l &\overset{\text{def.}}{=} \tilde{C}_l \setminus \bigcup_{s < l} \tilde{C}_s \\ \tilde{C}_l &\overset{\text{def.}}{=} \{x \in \mathbb{R}^n : \|x - p_l\| = \min_{s=1,\dots,\tilde{L}} \|x - p_s\|\}. \end{aligned} \tag{22}$$

*Let* $\mathcal{H}_L$ *be the set of functions from* $\mathbb{R}^n$ *to* $\{0,1\}$ *of the form*

$$f = \sum_{l=1}^{\tilde{L}} f_l \, I_{C_l}$$

*where* $\tilde{L} \in \mathbb{N}_+$ *with* $\tilde{L} \leq L$, $f_1, \dots, f_{\tilde{L}} \in \mathcal{H}$, *and* $(C_l)_{l=1}^{\tilde{L}} \in \mathcal{C}_L$. *Then, the VC dimension of* $\mathcal{H}_L$ *satisfies*

$$\mathrm{VC}(\mathcal{H}_L) \leq 8L \log(\max\{2, L\})^2 \big(\max\{d, 2(n+1)(L-1)\log(3L-3)\}\big).$$

*Proof of Lemma B.3.* Let us first fix our notation. Each $x_0 \in \mathbb{R}^n \setminus \{0\}$ and $t \in \mathbb{R}$ defines a *halfspace* in $\mathbb{R}^n$ given by $HS_{x_0,t} \overset{\text{def.}}{=} \{x \in \mathbb{R}^n : \langle x_0, x \rangle \leq t\}$ (see (Boyd & Vandenberghe, 2004, Section 2.2.1) for details). We denote set of all halfspaces in $\mathbb{R}^n$ by $\mathcal{HS}_n \overset{\text{def.}}{=} \{HS_{x_0,t} : \exists x_0 \in \mathbb{R}^n \setminus \{0\} \; \exists t \in \mathbb{R}\}$. Consider the set $\mathcal{C}(L)$ of all $C \subseteq \mathbb{R}^n$ of the form

$$C = \bigcap_{l=1}^{\tilde{L}-1} X_l \tag{23}$$

for some positive integer $2 \leq \tilde{L} \leq \max\{2, L\}$ and $X_1, \ldots, X_{\tilde{L}-1} \in \mathcal{HS}_n$.

**Step 1 - Reformulation as Set of Sets**
By definition of the powerset $2^{\mathbb{R}^n}$ of the set $\mathbb{R}^n$, each subset $A \subseteq \mathbb{R}^n$ can be identifies with a function (classifier) from $\mathbb{R}^n$ to $\{0,1\}$ via the bijection mapping any $X \in 2^{\mathbb{R}^n}$ to the binary classifier $I_X$ (i.e. the indicator function of the set $X$). Using this bijection, we henceforth identify both $\mathcal{H}$ and $\mathcal{H}_L$ with subsets of the powerset $2^{\mathbb{R}^n}$.

Under this identification, the class $\mathcal{H}_L$ can be represented as the collection of subsets $X$ of $\mathbb{R}^n$ of the form

$$X = \bigcup_{l=1}^{\tilde{L}} H_l \cap C_l, \tag{24}$$

where $\tilde{L} \in \mathbb{N}_+$ satisfies $\tilde{L} \leq L$, and for each $l = 1, \ldots, \tilde{L}$ we have $H_l \in \mathcal{H}$ and $(C_l)_{l=1}^{\tilde{L}} \in \mathcal{C}_L$ is of the form equation 22 for some *distinct* points $p_1, \ldots, p_{\tilde{L}} \in \mathbb{R}^n$.

**Step 2 - VC Dimension of Voronoi Diagrams with at-most $L$ Cells**
An element of $(C_l)_{l=1}^{\tilde{L}}$ of $\mathcal{C}_L$ is, by definition, a Voronoi diagram in $\mathbb{R}^n$ and thus, Boyd & Vandenberghe (2004, Exercise 2.9) implies that each $C_1, \ldots, C_{\tilde{L}}$ is the intersection of $\tilde{L}-1 \leq L-1$ halfspaces; i.e. $C_1, \ldots, C_{\tilde{L}} \in \mathcal{C}(L)$ (see equation 23). Since $\mathcal{C}_L = \{\cap_{l=1}^{\tilde{L}} H_i : \exists \tilde{L} \in \mathbb{N}_+ \; H_1, \ldots, H_{\tilde{L}} \in \mathcal{HS}_n \; \tilde{L} \leq L\}$ then Blumer et al. (1989, Lemma 3.2.3) implies that

$$\begin{aligned} \text{VC}(\mathcal{C}_L) \leq& 2\,\text{VC}(\mathcal{HS}_n)\,(L-1)\,\log(3L-3) \\ \leq& 2(n+1)\,(L-1)\,\log(3L-3); \end{aligned} \tag{25}$$

the second inequality in equation 25 holds since $\text{VC}(\mathcal{HS}_n) = n+1$ by Shalev-Shwartz & Ben-David (2014, Theorem 9.3).

**Step 3 - VC Dimension of The Class $\mathcal{H}_L$**
Define $\mathcal{H} \cap \mathcal{C}(L) \overset{\text{def.}}{=} \{H \cap C : H \in \mathcal{H} \text{ and } C \in \mathcal{C}(L)\}$. Again using Blumer et al. (1989, Lemma 3.2.3), we have

$$\begin{aligned} \text{VC}\big(\mathcal{H} \cap \mathcal{C}(L)\big) \leq& 2\,\big(\max\{\text{VC}(\mathcal{H}), \text{VC}(\mathcal{C}(L))\}\big)\,2\,\log(6) \\ \leq& 4\,\log(6)\,\big(\max\{\text{VC}(\mathcal{H}), \text{VC}(\mathcal{C}(L))\}\big) \\ \leq& 4\,\log(6)\,\big(\max\{d, 2(n+1)\,(L-1)\,\log(3L-3)\}\big). \end{aligned} \tag{26}$$

Consider the set $\tilde{\mathcal{H}} \overset{\text{def.}}{=} \{\cup_{l=1}^{\tilde{L}} H_l : \tilde{L} \in \mathbb{N}_+, \; \tilde{L} \leq L, \; \forall l = 1, \ldots, \tilde{L}, \; H_l \in \mathcal{H} \cap \mathcal{C}(L)\}$. Applying Blumer et al. (1989, Lemma 3.2.3), one final time yields

$$\text{VC}\big(\tilde{\mathcal{H}}\big) \leq 2\,\text{VC}\big(\mathcal{H} \cap \mathcal{C}(L)\big)\,L\,\log(3L) \tag{27}$$

$$\leq 2\Big(4\,\log(6)\,\big(\max\{d, 2(n+1)\,(L-1)\,\log(3L-3)\}\big)\Big)\,L\,\log(3L) \tag{28}$$

$$\leq 8L\,\log(\max\{2,L\})^2\,\big(\max\{d, 2(n+1)\,(L-1)\,\log(3L-3)\}\big) \tag{29}$$

where equation 28 held by the estimate in equation 26. Finally, since $\text{VC}(A) \leq \text{VC}(B)$ whenever $A \subseteq B$ for any set $B$ then since $\mathcal{H}_L \subseteq \tilde{\mathcal{H}}$ then equation 27-equation 29 yields the desired conclusion. □

We may now derive Theorem 4.2 by merging Lemma B.3 and one of the main results of Bartlett et al. (2019).

*Proof of Theorem 4.2.* Let $n, J, W, L \in \mathbb{N}_+$ and consider the (non-empty) set of real-valued functions $\mathcal{NN}_{J,W:n,1}^{\mathrm{PReLU}}$. By definition of the VC dimension of a set of real-valued functions, given circa equation 3, we have

$$\mathrm{VC}\left(\mathcal{NN}_{J,W:n,1}^{\mathrm{PReLU}}\right) \overset{\text{def.}}{=} \mathrm{VC}\left(I_{(0,\infty)} \circ \mathcal{NN}_{J,W:n,1}^{\mathrm{PReLU}}\right)$$
$$\mathrm{VC}\left(\mathcal{NP}_{J,W,L:n,1}^{\mathrm{PReLU}}\right) \overset{\text{def.}}{=} \mathrm{VC}\left(I_{(0,\infty)} \circ \mathcal{NP}_{J,W,L:n,1}^{\mathrm{PReLU}}\right). \tag{30}$$

By Bartlett et al. (2019, Theorem 7), we have that

$$\mathrm{VC}(I_{(0,\infty)} \circ \mathcal{NN}_{J,W:n,1}^{\mathrm{PReLU}}) \le D^\star \overset{\text{def.}}{=} \left\lceil J + (J+1)\, W^2 \, \log_2\left(e\, 4(J+1)\, W \log_2(e2(J+1)W)\right)\right\rceil. \tag{31}$$

Therefore, applying Lemma B.3 with $\mathcal{H} = \left(I_{(0,\infty)} \circ \mathcal{NN}_{J,W:n,1}^{\mathrm{PReLU}}\right)$ yields the estimate

$$\mathrm{VC}\left(I_{(0,\infty)} \circ \mathcal{NP}_{J,W,L:n,1}^{\mathrm{PReLU}}\right) \le 8L \log(\max\{2,L\})^2 \left(\max\{D^\star, 2(n+1)\,(L-1)\,\log(3L-3)\}\right) \tag{32}$$

Combining equation 32 and the definition equation 30 yields the bound. In particular,

$$\mathrm{VC}\left(\mathcal{NP}_{J,W,L:n,1}^{\mathrm{PReLU}}\right) \in \mathcal{O}\left(L \log(L)^2 \, \max\{nL \log(L), JW^2 \log(JW)\}\right)$$

yielding the second conclusion. $\qquad\square$

### B.3 Proof of Proposition 4.4

*Proof.* We argue by contradiction. Suppose that $\mathcal{F}$ has finite VC dimension $\mathrm{VC}(\mathcal{F})$. Then, Shen et al. (2022b, Theorem 2.4) implies that there exists a 1-Lipschitz map $f : [0,1]^n \to \mathbb{R}$ such that does not exist a *strictly positive* $\varepsilon \in (0, \mathrm{VC}(\mathcal{F})^{-1/n}/9)$ satisfying such that

$$\inf_{\hat{f} \in \mathcal{F}} \sup_{x \in [0,1]^n} |\hat{f}(x) - f(x)| \le \varepsilon. \tag{33}$$

However, Shen et al. (2022a, Theorem 1) implies that, for every 1-Lipschitz function, in particular for $f$, and for each $\tilde{\varepsilon} > 0$ there exists a $\hat{f}_{\tilde{\varepsilon}} \in \mathcal{F}$ satisfying

$$\sup_{x \in [0,1]^n} |\hat{f}_{\tilde{\varepsilon}}(x) - f(x)| \le \tilde{\varepsilon}. \tag{34}$$

Setting $\tilde{\varepsilon} = \mathrm{VC}(\mathcal{F})^{-1/n}/18$ yields a contradiction as equation 33 and equation 34 cannot both be simultaneously true. Therefore, $\mathcal{F}$ has infinite VC dimension. $\qquad\square$

## C The Curse of Irregularity

We now explain why learning Hölder functions of low regularity $((1, 1/d)$-Hölder) functions on the real line segment $[0, 1]$ is equally challenging as learning regular functions (1-Lipschitz) on $[0, 1]^n$.

### C.1 Hölder Functions

Fix $n, m \in \mathbb{N}$ and let $\mathcal{X} \subset \mathbb{R}^n$ be non-empty and compact of diameter $D$. Fix $0 < \alpha \le 1$, $L \ge 0$, and let $f : \mathcal{X} \to \mathbb{R}^m$ be $(\alpha, L)$-Hölder continuous, meaning

$$\|f(x) - f(y)\| \le L\|x - y\|^\alpha$$

holds for each $x, y \in \mathcal{X}$. For any $L \ge 0$ and $0 < \alpha \le 1$, we denote set of all $(\alpha, L)$-Hölder functions from $\mathcal{X}$ to $\mathbb{R}^n$ is denoted by $C^\alpha([0,1]^n, \mathbb{R}; L)$.

We focus on the class of locally Hölder functions since they are generic, i.e. universal, amongst all continuous functions by the Stone–Weierstrass theorem. In this case, Hölder functions are sufficiently rich to paint a

full picture of the hardness to approximate arbitrary Hölder functions either by MLPs against the proposed model.

In contrast to smaller generic function classes, such as polynomials, Hölder functions provide more freedom in experimentally visualizing our theoretical results. This degree of freedom is the parameter $\alpha$, which modulates their *regularity*. As $\alpha$ tends to 0 the Hölder functions become complicated and when $\alpha = 1$ the Rademacher-Stephanov theorem, see Federer (1978, Theorems 3.1.6 and 3.1.9) characterizes Lebesgue almost-everywhere differentiable functions as locally $(1, L)$-Hölder maps. Note that $(1, L)$-Hölder functions are also called $L$-Lipschitz maps and, in this case, $L = \sup_x \|\nabla f(x)\|_{op}$ where the supremum is taken over all points where $f$ is differentiable and where $\|\cdot\|_{op}$ is the operator norm.

## C.2 The Curse of Irregularity

The effect of *low Hölder regularity*, i.e. when $\alpha \approx 0$, has the same effect as high-dimensionality on the approximability of arbitrary $\alpha$-Hölder functions. This is because any real-valued model/hypothesis class $\mathcal{F}_1$ of functions on $\mathbb{R}$ approximating an arbitrary $(\frac{1}{d}, 1)$-Hölder functions By Shen et al. (2022b, Theorem 2.4), we have the lower minimax bound: if for each $\varepsilon > 0$ we have *"the curse of irregularity"*

$$\sup_f \inf_{\hat{f} \in \mathcal{F}_1} \sup_{0 \leq x \leq 1} |f(x) - \hat{f}(x)| \leq \varepsilon \Rightarrow \text{VC}(\mathcal{F}_1) \in \Omega(\varepsilon^{-d}) \tag{35}$$

where the supremum is taken over all $f \in C^{1/d}([0, 1], \mathbb{R}; 1)$. The familiar curse of dimensionality also expresses the hardness to approximate an arbitrary 1-Lipschitz $((1, 1)$-Hölder), thus relatively regular, function on $[0, 1]^n$. As above, consider any model/hypothesis class $\mathcal{F}_2$ of real-valued maps on $\mathbb{R}^d$ then, again using Shen et al. (2022b, Theorem 2.4), one has the lower-bound

$$\sup_f \inf_{\hat{f} \in \mathcal{F}_2} \sup_{x \in [0,1]^n} \|f(x) - \hat{f}(x)\| \leq \varepsilon \Rightarrow \text{VC}(\mathcal{F}_2) \in \Omega(\varepsilon^{-d}) \tag{36}$$

where the supremum is taken over all $f \in C^1([0, 1]^n, \mathbb{R}; 1)$. Comparing equation 35 and equation 36, we find that the difficulty of uniformly approximating an arbitrary *low-regularity* $((\frac{1}{d}, 1)$-Hölder) function on a 1-dimensional domain is roughly just as complicated as approximating a relatively regular (1-Lipschitz) function on a *high-dimensional* domain.

Incorporating these lower bounds with the lower-bound in equation 4, we infer that the minimum number layers ($L$) and minimal width ($W$) of each MLP approximating a low-regularity function on the low-dimensional domain $[0, 1]$ is roughly the same as the minimal number of layers and width of an MLP approximating a high-regularity map on the high-dimensional domain $[0, 1]^n$.

# D Experimental details

We include here experimental details, we refer to the source code in the supplementary material for more details. We first outline the algorithm used to train the MoMLP MoE model. We then provide details on the trained architecture and hyperparameter details in the implementation.

## D.1 Definitions of the Ackley and Rastrigin functions

Let us note $x = (x_1, \ldots, x_n)^\top \in \mathbb{R}^n$ the $n$-dimensional representation of a sample, we use the following formulation of the Ackley function:

$$\text{Ackley}(x) = 20 + \exp(1) - a \exp\left(-b\sqrt{\frac{1}{n}\sum_{i=1}^n x_i^2}\right) - \exp\left(\frac{1}{n}\sum_{i=1}^n \cos(2\pi x_i)\right) \tag{37}$$

where $a = 20$ and $b = 0.2$. We also use the following formulation of the Rastrigin function:

$$\text{Rastrigin}(x) = \sum_{i=1}^n x_i^2 + 10\left(n - \sum_{i=1}^n \cos(2\pi x_i)\right) \tag{38}$$

## D.2 Training Algorithm

We now provide an explanation for the training algorithm. As discussed, we mitigate down the algorithm into two parts: discovering the prototypes and training the networks. Conceptually, the prototypes define where in the input space the networks are located, or in other words, where in the input space we expect each of the networks to have the best performance. During inference, we will route a given input to the appropriate network based on its nearest prototype. In essence, each network learns to approximate a specific region of the overall input domain.

**Discovering prototypes.** In principle, we may not know how to partition the input space. One approach is to utilize standard clustering algorithms like $K$-means, but this might be suboptimal for the downstream task unless we are already operating in a structured latent space, such as those found in pre-trained models (further discussion on this is available in Section 6). Another way is to learn it via gradient descent by optimizing the location of the prototypes for a specific task by following the gradient of the downstream loss. At the beginning of training, we have $\hat{F}(x) \stackrel{\text{def.}}{=} (\hat{f}_1(x), \ldots, \hat{f}_K(x))$ which contains a collection of $K$ randomly initialized shallow or small networks (i.e., much smaller than our MoMLPs described later). In this first step, we assume that we are able to load all randomly initialized networks into our GPU memory. In particular, this is true because we use small networks with few parameters, which we will later "deepen" in the next step by adding additional hidden layers. We initialize the prototypes $p \stackrel{\text{def.}}{=} (p_1, \ldots, p_K)$ randomly from a uniform distribution within the bounds of our training dataset input samples. We use the following expression to train the location of our prototypes $\{p_k\}_{k=1}^{K}$ in $\mathbb{R}^n$ by minimizing the energy:

$$\sum_{(x,y)\in\mathcal{D}} \ell\big( \text{softmax}\big( -\|x - p_i\|_{i=1}^{K}\big)^{\top} \hat{F}(x), y\big). \tag{39}$$

where the loss $\ell$ is task-specific; for example, one could use mean squared error for regression and cross-entropy for classification. The softmax weights the importance of the prediction of each of the MoMLPs in $\hat{F}$ for a given input $x$, as a function of the input's distance to the prototypes, $\|x - p_i\|_{i=1}^{K} \stackrel{\text{def.}}{=} (\|x - p_1\|, \ldots, \|x - p_K\|)$. Both the locations of the prototypes and the shallow randomly initialized neural networks assigned to them are optimized.

**Deepening the Networks.** After the initial training phase, we enhance the networks by incorporating additional layers. Specifically, we introduce linear layers with weights initialized to the identity matrix and bias set to zero, just before the final output layer of each network. To encourage gradient flow in these new layers during the subsequent training stage, we slightly perturb this initialization with small Gaussian noise. This approach is driven by the fact that in the second training stage, each MoMLP can be optimized in a distributed manner. Consequently, we can work with larger networks without the need to load all of them simultaneously into our GPU, allowing for more model parameters. During the first stage, we have already optimized our networks alongside the prototype locations, converging towards a minimum. By initializing the networks with the new layers close to the identity, we can ensure that their output at the start of the second stage of training is similar to that produced by the original networks. This allows us to smoothly continue the optimization process from the point where we previously halted.

**MoMLP Training.** Once prototype locations have been fixed we can independently train MoMLPs $\hat{f}_1, \ldots, \hat{f}_K$ by minimizing for all $k \in \{1, \ldots, K\}$:

$$\sum_{\substack{(x,y)\in\mathcal{D} \\ k\in\arg\min_{j\in\{1,\ldots,K\}}\{\|x-p_j\|\}}} \ell(\hat{f}_k(x), y) \tag{40}$$

over all the networks. We optimize the MoMLP network $\hat{f}_k$ for training data points that are closest to prototype $p_k$. The training procedure is summarized in Algorithm 2.

**Inference.** At inference time, each test sample $x$ is assigned to its nearest prototype $p_k$ where $k \in \arg\min_{j\in\{1,\ldots,K\}}\{\|x - p_j\|\}$ and the prediction is made by the $k$-th MoMLP $\hat{f}_k$.

**Comparison to Standard Distributed Training.** One can distribute the complexity of feedforward models by storing each of their layers in offline memory and then loading them sequentially into VRAM

---

**Algorithm 2:** MoMLPs Training.

---

**Require:** Training data $\mathcal{D} \stackrel{\text{def.}}{=} \{(x_j, y_j)\}_{j=1}^{N}$, no. of prototypes $K \in \mathbb{N}_+$, loss function $\ell$.

    **Discovering Prototypes:**

$$(\hat{F}, p) \leftarrow \underset{\hat{F}, p}{\arg\min} \sum_{(x,y) \in \mathcal{D}} \ell\left(\text{softmax}\left(x|p\right)^\top \hat{F}(x), y\right)$$

    **Deepen networks:**

        **For** $k = 1, \ldots, K$**:**

        $\hat{f}_k \leftarrow \text{deepen}(\hat{f}_k)$

    **MoMLP Training:**

        **For** $k = 1, \ldots, K$**:**

$$\hat{f}_k \leftarrow \underset{\hat{f}_k}{\arg\min} \sum_{\substack{(x,y) \in \mathcal{D} \\ k \in \arg\min_j \{\|x - p_j\|\}}} \ell(\hat{f}_k(x), y)$$

    **return** MoMLP parameters $\{\hat{f}_k\}_{k=1}^{K}$ and prototype locations $\{p_k\}_{k=1}^{K}$.

---

during the forward pass. This does avoid loading more than $\mathcal{O}(\text{Width})$ active parameters into VRAM at any given time, where Width denotes the width of the feedforward model. However, doing so implies that all the model parameters are ultimately loaded during the forward pass. This contrasts with the MoMLP model, which requires $\mathcal{O}\left(\log_2(K)\, \text{Width}^2\, \text{Depth}\right)$ to be loaded into memory during a forward pass; where Width and Depth are respectively the largest width and depth of the MLP at any leaf of the tree defining a given MoMLP, and $K$ denotes the number of prototypes. However, in the forward pass, one loads $\mathcal{O}(\varepsilon^{-n/2})$ parameters for the best worst-case MLP while only $\mathcal{O}(n \log(1/\varepsilon)/\varepsilon)$ are needed in the case of the MoMLPs. The number of parameters here represents the optimal worst-case rates for both models (see Table 8 and Theorem 5.3).

## D.3 Architectures and hyperparameters

Lastly, we detail the model architectures and hyperparameters used in our experiments.

### D.3.1 MoMLPs

We set the width of our MoMLPs to $w = 1000$. In other words, each hidden layer of our MoMLPs contains a linear matrix of size $w \times w$.

In the regression task, our MoMLPs contain 3 hidden layers and we use a learnable PReLU as activation function. For training, we use the Adam (Kingma & Ba, 2014) optimizer with a learning rate of $10^{-4}$ and the default Pytorch hyperparameters.

In the classification task, we follow the setup of Oquab et al. (2023) and use AdamW (Loshchilov & Hutter, 2019) as the optimizer with a learning rate of $10^{-3}$ and the default parameters from PyTorch. Our MoMLPs consist of four hidden layers for the classification task, and we apply BatchNorm1d before the PReLU activation function.

### D.3.2 Baseline

The baseline shares the same architecture as the MoMLPs described above. However, if we let $K$ denote the number of prototypes and assume that $\sqrt{K}$ is a natural number, the width of the baseline is $w\sqrt{K}$, ensuring that the total number of hidden parameters matches that of all the MoMLPs combined. In our experiments, we set $K = 4$, so $\sqrt{K} = 2$.

## D.4 Additional classification experiment

Instead of using the DINOv2 features as input to our model, we also conducted an experiment where the $32 \times 32$ CIFAR RGB images were vectorized into 3072-dimensional vectors, which were then used as input to

our model (recall that $32 \times 32 \times 3 = 3072$). The elements of these vectors were normalized to lie within the interval [0,1].

The accuracy scores, reported in Table 9, demonstrate that the performance of our approach is comparable to that of the baseline while requiring less VRAM per expert. It is important to note, however, that since we are using standard MLPs, the reported scores are lower than those achievable with neural networks specifically designed for computer vision tasks.

*Table 9:* Test classification accuracy using the vectorized RGB images as input (average and standard deviation).

| Dataset | CIFAR-10 | CIFAR-100 |
|---|---|---|
| Ours (weighted) | $46.79 \pm 0.91$ | $20.82 \pm 0.29$ |
| Ours (unweighted) | $46.09 \pm 0.74$ | $20.62 \pm 0.28$ |
| Baseline | $47.36 \pm 1.58$ | $19.01 \pm 3.42$ |

