# OpenReview forum: "Approximation Rates and VC-Dimension Bounds for (P)ReLU MLP Mixture of Experts"
_TMLR — Accepted by TMLR_

### Review · Reviewer_y48i · 2024-10-29

**Summary Of Contributions:**

This paper provides a theoretical analysis of a mixture of MLP models in approximating any Lipschitz function. In particular, the authors theoretically analyze the trade between No. experts needed and the complexity of each expert. Such analysis is critical for understanding the practicality of large-scale MoE models. As described in the paper, it can be beneficial to maintain a low complexity for each model during inference. Therefore, it is crucial to demystify the overall MoE scale with each model's size and objective functions.

**Audience:**

Yes

**Claims And Evidence:**

Yes

**Requested Changes:**

1 Include the Lipschitz parameter in the analysis, or clearly explain why it can be ignored.

2 More explanations on $r$, and how it connects to the actual objective function or the MoE models to be analyzed

**Strengths And Weaknesses:**

**Strengths:**

The authors provide a valuable analysis of the tradeoff between No. experts and the complexity of each expert. Such analysis is very important not only for theoretically understanding MoE models, but for practically designing large-scale MoE models.

---

**Weakness:**

1 The analysis does not consider how the Lipschitz parameter affects the tradeoff between No. expert and expert complexity

It is odd that Theorem 4.1 did not consider the Lipschitz parameter as a factor in affecting the tradeoff between No. expert and expert complexity. The target objective function should play a role in the analysis. However, I believe it is missing in the current version.

2 The introduction of $r$ lacks sufficient explanation

I am confused about the role of $r$ in the analysis. The authors need to provide a clear explanation regarding the "number of experts-to-expert complexity trade-off parameter." It is hard to interpret and understand what this parameter is describing.

2.1 What is $r = 0$?

According to the conclusion, such as in Table 1, if $r=0$, the number of parameters per expert will be 1. However, it is unclear what case corresponds to $r=1$, such as what the objective function will be when $r=0$. Can the authors provide more explanations on this?

---

> ### Author Response · Authors · 2024-10-29
> **RE**
>
> Dear reviewer,
>
> Thank you for taking time to review our paper and for your kind feedback.
>
> 1. Table 1 provides order estimates, the exact bounds are given in Appendix A, were we spell out the explicit dependance on the modulus of continuity (and therefore Lipschitz constant in the special Lipschitz case) on each MoE parameter: depth, width, and number of expects.
> We can make the link to the (rather technical) Tables 4 and 6 more explicit if you would like.
>
> 2. When r=0, there is only a single expert (see Table 4 for example).
>
> 3. This is incorrect, the number will be constant O(1) not 1.
>
> We are a bit unclear on what you mean by "the objective function", as it stands, we are only discussing approximation power with respect to the (standard) $\ell^2$ norm.  Can you provide more detail on what you mean by the second portion of question 3?

---

### Review · Reviewer_imma · 2024-11-07

**Summary Of Contributions:**

This paper provides an approximation and learning theoretic analysis of mixtures of experts MLPs with trainable \(P\)ReLU activations. They prove that any Lipschitz function on [0,1] can be approximated uniformly, arbitrarily well with MoML]
P requiring only $O(\epsilon^{-1})$ parameters. The also provide generalization bounds based on the VC dimension of the MoMLP models.

**Audience:**

Yes

**Claims And Evidence:**

Yes

**Requested Changes:**

- Can you clarify what $\mathcal{K}$ is in the statement of Theorem 4.1.
- In the statement of Theorem 4.1 is the activation function learnable and does the $\alpha$ in the activation function correspond to the $\alpha$ of the $\alpha$-Holder function being approximated?
- I would suggest moving some of th empirical results from Apendix D to demonstrate how the theory connects to practice. Even in this toy experiments.

**Strengths And Weaknesses:**

### Strengths
- To the best of my knowledge this is one of the few papers studying the theoretical foundations of MoE models. As this model is becoming more common with larger deep learning systems this theory can be used to influence the design and implementation of MoE models.
- The paper is clear and well written.

### Weaknesses
- The only weakness of the paper is the simplicity of the setting considered. Although this is somewhat okay since this is taking the first theoretical step to understand mixtures of MLPs.

---

> ### Author Response · Authors · 2024-11-28
>
> We thank the reviewer for the comments.
>
> ## Request 1
>
> We thank you for noticing this typo.  In the technical generalization of Theorem 4.1, recorded in Theorem 5.3, $\mathcal{K}$ is allowed to be any non-empty compact subset of the domain.  However, in this simplified formulation, it should be $\bar{B}_n(0,1)$.  We have both corrected this typo, and added a remark just below the result in Theorem 4.1, briefly previewing how it can, and is, extended in Theorem 5.3 below.
>
> ## Request 2
>
> We thank you for noting that the choice of symbols for the trainable activation function is sub-optimal as we were also using α for the Hölder exponent.  We have improved our notation, using γ for the trainable activation function parameter; reserving α only for the Hölder exponent.  That said, *yes* the activation function parameter γ is indeed trainable, and *no* it does not depend on the Hölder coefficient.
>
> ## Request 3
>
> We have moved the empirical results to the main paper as suggested, it is section 6 now.
>
> All changes are highlighted in green. Please let us know if you have any further questions and we will be happy to address them.

---

### Review · Reviewer_pxj4 · 2024-11-21

**Summary Of Contributions:**

This paper looks at VC dimension  and approximation rates of Mixture of Experts MLPs with (P)ReLU activations through the prism of data prototypes. One of the most interesting results presented in the paper, in my opinion, is that the approximation quality degrades with the large number of small experts. A small number of heavy experts seems to be theoretically justified.

**Audience:**

Yes

**Broader Impact Concerns:**

This work has no ethical concerns.

**Claims And Evidence:**

Yes

**Requested Changes:**

1. I suggest that the authors move experimental results into the main paper, and show qualitatively how their theoretical results related to approximation quality and the number of experts hold in practice. This experiment should not use any DINO embeddings. For such small data as CIFAR, a convent + regular neural net should (training from scratch).

2. Another important request that I have, is to address the logical flaw between the claims and the results.

3. Add statistical analysis when assessing results (e.g. models in Table 8 have no statistical difference)

**Strengths And Weaknesses:**

## Strengths
The biggest strength of this paper is the clarity of narrative, as well as great teaching outcomes. I learned a lot. The authors rigorously looked at several aspects of the MoMLPs, and provided great theoretical contributions. While they have rather small scope, they are still of interest to TMLR audience.

## Weaknesses
1. The biggest concern about this paper is the lack of practical utility. At least an empirical study verifying the claims of approximation result would be great. I would encourage authors to work w/o DINO embeddings, and show that the results actually hold w/o pre-training. On CIFAR, DINO embeddings are very strong due to ImageNet pretraining.

2. Another weakness of the paper is that the authors claim mixture of experts bringing benefit because one does not need to load the whole set of experts into memory. At the same time, their results (as well as the empirical results from other works in the field) show that one does not need to have many experts. This is a logical flaw, and it has to be fixed.

3. Statistical analysis of the results is  missing

---

> ### Author Response · Authors · 2024-11-28
>
> We thank the reviewer for the insights. We have uploaded a revised version of the paper and highlighted changes in green.
>
> ## Weakness 1: Practical Utility and DINO embeddings
>
> In terms of practical utility, the idea of decomposing a single large neural network into multiple smaller models that are run in parallel on different clusters/machines/GPUs has been successfully used in computer vision tasks such as refs [A,B] (and also several other references which can be found in Section 1 of the paper). However, these works focus on applications, and they do not provide theoretical approximation rates like our work does. **The aim of our work is to provide theoretical foundations for an empirical phenomena that has already been observed by practitioners.** We believe our framework is a first theoretical step in that direction. We also believe refs [A,B] are other good illustrations of practical results of this kind of idea beyond the experiments presented in our paper.
>
> Our theoretical work focuses on MLPs, which is why we utilize DINO embeddings for the image classification task. This approach allows us to extract embeddings that can be fed into an MLP for classification in computer vision tasks. Otherwise, we would need to use other architectures, such as CNNs, which are not covered by our theoretical guarantees, although they represent a potential direction for future theoretical research. In summary, **we work with data representations that can be readily processed by an MLP, ensuring close adherence to the theory**. While it is true that DINO embeddings already serve as good representations of images due to pretraining, the goal of our work is to compare the relative performance of a single expert to that of multiple experts. Since all configurations (single large MLP vs MoMLPs) are evaluated using the same input-output pairs, the comparison is fair, as all models benefit equally from the DINO features.
>
> [A] Ren et al. Xcube: Large-scale 3d generative modeling using sparse voxel hierarchies
> [B] Song et al. Multi-student diffusion distillation for better one-step generators
>
> ## Suggestion 1: Experimental results in the main text
>
> Following your suggestion, we have moved most of the experimental section in the Appendix to the main paper (while staying within the 12 page limit). We have also added in Section 6.3 a short discussion about some points that seemed unclear.
>
> ## Weakness 2 and Suggestion 2: Statement Logic
>
> We would like to clarify that our result does not claim that MoEs only benefit when large numbers of experts are used, as one can vary the ratio of expert-to-expert complexity hyperparameter “r”.  Rather, we claim that the more experts used the smaller each expert needs to be in the worst-case.  Therefore, if one, theoretically, has access to a massive number of experts, then each expert can be extremely simple in theory.
>
> We agree that in practice, one often does not need to utilize a massive number of experts to achieve good predictive power of the MoE.  However our results should not be interpreted as best-case results; rather, as they are universal approximation theorems, they are worst-case guarantees.  In this way, our results imply that a massive number of experts allows each expert to be smooth, no matter how difficult the approximation problem.
>
> In conclusion, **there is no logical flaw in our statement, but rather we discuss worst-case scenarios**. For instance, in some of our experiments we use as little as 4 experts. In practice, the optimal number of experts to be used is a complex tradeoff between the GPU memory available and the complexity of the underlying function we are trying to learn. If we can train relatively large experts, there is no point in having a multitude of really simple and shallow experts that do not properly use the compute capabilities of our GPUs.
>
> ## Weakness 3 and Suggestion 3: Statistical significance of the results
>
> Regarding statistical difference, before directly addressing the concern of the reviewer we would like to highlight that all results are presented as the average results across 10 different runs and we also report the standard deviation. This is the case both for regression and classification experiments.
>
> Concerning the statistical analysis in the classification task (Table 5 in the updated manuscript), **the primary goal of our experiments was to demonstrate that our MoMLPs can match** (or sometimes outperform) **the performance of a single large neural network**. This is particularly advantageous in scenarios where a single large neural network cannot fit into GPU. We acknowledge that there was no significant difference in accuracy between our MoMLPs and the baseline, but this aligns with our objective: each expert model requires less memory and can run in parallel while maintaining the performance of the large model.
>
> Following the advice of the reviewer, we have removed the bold from the table as there is no statistical significance.

---

> > ### Comment · Reviewer_pxj4 · 2024-12-05
> > **Weakness 1: Practical Utility and DINO embeddings**
> >
> > The point I am making about DINO, is that they basically create embedding space, where one can fit a linear model to get a good classificator. I thus suggested the authors to go directly to raw pixel data. The images from the used datasets are anyway small, and would fit straight to memory. This will make the results validating theory much more convincing. It would also eliminate all potential concerns from the readers.

---

> > > ### Author Response · Authors · 2024-12-11
> > > **Weakness 1: Practical Utility and DINO embeddings**
> > >
> > > Thank you for your suggestion. We have included an experiment in Appendix D.4 where, instead of using Dino features, we vectorize the 32x32 RGB images from the CIFAR dataset into 3072-dimensional vectors, which are then used as inputs to our model. The conclusion remains the same: the performance of our approach is comparable to that of a large neural network while requiring less VRAM per expert. Since we use standard MLPs, it is important to note that the reported scores are lower than those achievable with neural networks designed for computer vision tasks, such as convolutional networks.

---

> > ### Comment · Reviewer_pxj4 · 2024-12-05
> > **Weakness 2 and Suggestion 2: Statement Logic**
> >
> > I would anticipate that the authors take an action, and revise text accordingly, so that there is unambiguous interpretation by the readers.

---

> > > ### Author Response · Authors · 2024-12-11
> > > **Weakness 2 and Suggestion 2: Statement Logic**
> > >
> > > Dear Reviewer,
> > >
> > > We now understand your point, which we had previously misunderstood.
> > >
> > > Following your advice, we have updated the text and added clarifications on page 2, below Table 1, as well as on page 7 in the discussion of Section 4.1. All changes for this second round of replies are highlighted in red. We hope this helps make the text clearer and less ambiguous for readers.

---

> > ### Comment · Reviewer_pxj4 · 2024-12-05
> > **Weakness 3 and Suggestion 3: Statistical significance of the results**
> >
> > Great! Thanks for resolving this issue.

---

### Author Response · Authors · 2025-01-08
**Thanks!**

Dear Reviewers,

We are grateful for your helpful feedback and the positive discussion around our paper, which we have integrated into our revised manuscript.  We eagerly look forward to your recommendations.

Best,
The authors

---

### Decision · Action_Editor_emKN · 2025-01-22

**Recommendation:** Accept as is

**Comment:**

The consensus is quite positive on this paper. All reviewers note the redactional quality as well as the interesting results on a rather new field of study. There is some slight concern concerning the quality of the experiments or the actual practicality of the results. But given the theoretical orientation of the paper, this is not a major issue. However one reviewer still has issues with some seemingly important definition (r); maybe the authors can have a last thought on this before final publication.
Therefore, given the overall positive comments by the reviewers on this paper that looks at a young subject, I recommend publication.

**Audience:**

All reviewers agree on the TMLR audience being relevant. The theoretical study of MoE seems rather new so the community may be small at the moment, but this is not a negative aspect, quite the opposite in my opinion.

**Claims And Evidence:**

All reviewers agree that both claims are well supported by evidence.